# Ethane-oxidising archaea couple $CO_2$ generation to $F_{420}$ reduction

Olivier N. Lemaire[1], Gunter Wegener[1,2,3] & Tristan Wagner [1,4] ✉

The anaerobic oxidation of alkanes is a microbial process that mitigates the flux of hydrocarbon seeps into the oceans. In marine archaea, the process depends on sulphate-reducing bacterial partners to exhaust electrons, and it is generally assumed that the archaeal $CO_2$-forming enzymes (CO dehydrogenase and formylmethanofuran dehydrogenase) are coupled to ferredoxin reduction. Here, we study the molecular basis of the $CO_2$-generating steps of anaerobic ethane oxidation by characterising native enzymes of the thermophile *Candidatus* Ethanoperedens thermophilum obtained from microbial enrichment. We perform biochemical assays and solve crystal structures of the CO dehydrogenase and formylmethanofuran dehydrogenase complexes, showing that both enzymes deliver electrons to the $F_{420}$ cofactor. Both multi-metalloenzyme harbour electronic bridges connecting CO and formylmethanofuran oxidation centres to a bound flavin-dependent $F_{420}$ reductase. Accordingly, both systems exhibit robust coupled $F_{420}$-reductase activities, which are not detected in the cell extract of related methanogens and anaerobic methane oxidisers. Based on the crystal structures, enzymatic activities, and metagenome mining, we propose a model in which the catabolic oxidising steps would wire electron delivery to $F_{420}$ in this organism. Via this specific adaptation, the indirect electron transfer from reduced $F_{420}$ to the sulphate-reducing partner would fuel energy conservation and represent the driving force of ethanotrophy.

Alkanes are the most reduced carbon compounds available in nature that can be used as cellular energy sources for microorganisms in oxic and anoxic environments[1–3]. Alkanes naturally perfuse in marine cold seeps and hydrothermal vents, but a biological filter composed of aerobic and anaerobic alkane-oxidising microorganisms prevents the alkanes release into the oceans and atmosphere while sustaining the surrounding chemoautotrophic microorganisms through sulphide generation[4–6]. The microorganisms performing the anaerobic oxidation of alkanes and their metabolisms are, however, relatively uncharacterised. The two ethane oxidisers are part of the *Methanosarcinales* order and were shown to catalyse the complete anaerobic oxidation of ethane, the second most abundant alkane in seeps[7–9].

These organisms are closely related and are proposed to be part of the same archaeal genus according to the Genome Taxonomy Data Base[10]. Ethane is activated as an ethyl-thiol adduct on the coenzyme M (CoM) via the ethyl-CoM reductase (ECR), an enzyme specific to these ethane-oxidising archaea[7,8,11]. It has been suggested that the generated ethyl-CoM is further processed to acetyl-Coenzyme A (acetyl-CoA) based on the knowledge acquired on methanogens belonging to the same order, together and supported by transcriptomics and proteomics data[7,8]. Based on the accepted metabolic model, the acetyl-CoA decarbonylase/synthase complex (ACDS) would transform the acetyl-CoA to generate $CO_2$ concomitantly with a methyl group branched on a tetrahydromethanopterin carrier ($CH_3$-$H_4$MPT, Fig. 1a) and CoA. The

[1]Max Planck Institute for Marine Microbiology, Celsiusstrasse 1, 28359 Bremen, Germany. [2]MARUM, Center for Marine Environmental Sciences, University of Bremen, Bremen, Germany. [3]Alfred Wegener Institute Helmholtz Center for Polar and Marine Research, Bremerhaven, Germany. [4]Present address: Institut de Biologie Structurale, 71 avenue des Martyrs, 38000 Grenoble, France. ✉e-mail: twagner@mpi-bremen.de

methyl group would be oxidised through the reverse methanogenesis pathway and released as $CO_2$ by the formylmethanofuran dehydrogenase complex (Fmd/Fwd for molybdenum/tungsten-dependent enzymes, respectively. Fig. 1a)[12–14]. Therefore, ethane would be ultimately oxidised into two molecules of $CO_2$, and the $CO_2$-releasing enzymes (ACDS and Fwd/Fmd complexes) are expected to reduce ferredoxin, which is employed for energy conservation in methanogens[15]. The electrons released during ethane oxidation are supposed to be indirectly or physically transferred to sulphate-reducing bacteria living in a syntrophic partnership with the archaea[7,8].

The ethanotrophs do not contain any known membranous systems that would allow energy conservation from ferredoxin oxidation, questioning if the $CO_2$-releasing step operated by ACDS and Fwd/Fmd would be necessarily coupled to ferredoxin reduction, as it is commonly assumed for methanogens and alkanotrophs. To solve this metabolic puzzle, we here characterise the multi-enzymatic ACDS and Fwd/Fmd complexes[16–23] by isolating the CODH component of the ACDS and the entire Fwd complex directly from a microbial enrichment of a syntrophic consortium composed of the ethane-oxidising archaeon *Candidatus* Ethanoperedens thermophilum and the sulphate-reducing bacterium *Candidatus* Desulfofervidus auxilii. The archaeon represents around 40% of the microbial population in the culture[8]. Purifying enzymes from such a heterogeneous microbial mixture is only feasible for highly abundant enzymes. Published transcriptomic data confirmed that the genes coding for the subunits of the ACDS and Fwd/Fmd complexes are among the 250 most expressed genes in the culture conditions[8]. The biochemical and structural characterisation of both complexes, as well as enzymatic assays, support that the $CO_2$-generating steps are coupled to $F_{420}$ reduction instead of ferredoxin, suggesting that $F_{420}$ reduction is the main driver of this metabolism[24].

## Results

### The isolated CODH component of ACDS and the Fwd/Fmd complex differ from characterised homologues

We anaerobically purified the native ACDS and Fwd/Fmd complexes of *Ca*. E. thermophilum, by multi-step chromatography from the enriched culture biomass (Fig. 1b, c). During the purification process, the ACDS was followed by measuring the viologen-dependent CO-oxidation activity of its CO-dehydrogenase (CODH) module. The Fwd/Fmd complex was followed by activity measurements monitoring the viologen-dependent oxidation of furfurylformamide, a surrogate of formylmethanofuran (CHO-MFR, Supplementary Table 1)[25]. Previous works showed that some Fwd/Fmd complexes could catalyse the oxidation of formate, albeit at low rates and high formate concentrations[26]. Hence, formate dehydrogenase activity was also monitored (Fig. 1c and Supplementary Table 1). We also tested acetate as a putative C2 substrate, but it yielded no detectable activity for the purified enzyme.

The enzymes with CO oxidation and furfurylformamide oxidation activities were anaerobically enriched from the microbial enrichment through a five and four-step purification protocol, respectively (Supplementary Table 1). The molecular weights of both purified complexes were estimated based on native PolyAcrylamide Gel Electrophoresis (PAGE) and size-exclusion chromatography (Fig. 1b, c and Supplementary Fig. 1). The molecular weight of the purified CODH complex of *Ca*. E. thermophilum was experimentally estimated at around 310 kDa. In *Methanosarcina* species, the CODH is part of the 2.4 MDa ACDS composed of five subunits ($\alpha_8\beta_8\gamma_8\delta_8\epsilon_8$ stoichiometry) or in an $\alpha_2\epsilon_2$ subcomplex with a molecular weight of 215 kDa[16,17]. The results hence indicate that the CO-oxidising enzyme of *Ca*. E. thermophilum have a different organisation than the previously described complex (Fig. 1b, Supplementary Fig. 1a, b). The experimentally determined molecular weight of the Fmd/Fwd complex from *Ca*. E. thermophilum (ranging from 174 to 253 kDa, Fig. 1c and

Supplementary Fig. 1c, d) also appears to be incoherent with the previously described complexes[19,20,22]. The compositions of both atypical complexes were elucidated through crystallographic snapshots.

### Structure and composition of the CODH subcomplex

The structure of the CODH component of the ACDS was obtained by X-ray crystallography and refined to a 1.89-Å resolution (Fig. 2a and Supplementary Table 2). As reflected by the denaturing PAGE profile (Fig. 1b), the complex organises as a dimer of three subunits: the $\alpha$ and $\epsilon$ subunits (respectively CAD7772032 and CAD7772037), already described in methanogens, and an additional subunit homologous to the $F_{420}H_2$-oxidase domain of the sulphite reductase from *Methanocaldococcus jannaschii*[27] (Fig. 2a and Supplementary Figs. 2–4). Because of its tight binding on the CODH component and its presence in the operon coding for the ACDS, this additional subunit will be referred to as $\zeta$ subunit (CAD7772047). Unless stated otherwise, the name CODH component will systematically refer to the $\alpha_2\epsilon_2\zeta_2$ complex below. All three subunits were detected by mass spectrometry analysis from the band exhibiting CODH activity on native PAGE, while no peptide belonging to the three other ACDS subunits ($\beta$, $\delta$ and $\gamma$) could be detected in this sample (Supplementary Table 3). The purified CODH complex is supposed to be a subcomponent of the ACDS for the following reasons: (i) the $\alpha$ and $\epsilon$ subunits are the sole isoforms encoded in the *Ca*. E. thermophilum genome, (ii) the genes encoding for the CODH component subunits are in the same genomic region encoding the other ACDS subunits (Fig. 2a) and (iii) genes coding for the ACDS complex are expressed in these culture conditions, and the enzyme is supposed to be part of the ethanotrophy pathway[7,8]. We suppose that the whole ACDS complex was destabilised upon cell lysis (e.g. due to the low affinity of the component subunits), simplifying the characterisation of the CODH component.

The $\alpha_2\epsilon_2$ core of the CODH component can be reliably superposed on the homologous structure from *Methanosarcina barkeri*[17], also obtained from dissociation from the ACDS complex (Supplementary Fig. 3a, b). The complete chains could be modelled except for the flexible N-terminal part of the $\alpha$ subunit (1–30, Supplementary Fig. 5) and a few residues at its C-terminal extremity. While the $\alpha$ subunit exhibits high structural conservation with the homologue from *M. barkeri*, the $\epsilon$ subunit slightly differs due to local reorganisations to accommodate the $\zeta$ subunit (Supplementary Fig. 3c). Each $\alpha/\epsilon$ interface of the complex from *Ca*. E. thermophilum accommodates an ion in close proximity to the [4Fe-4S] cluster 3 (modelled as a K atom, Supplementary Fig. 3d). In the structure of *M. barkeri*, the respective position is empty, suggesting that the additional ion may be a thermophilic adaptation of the enzyme to optimise the stability between the subunits (Supplementary Fig. 3d). All metallo-cofactors harboured on the $\alpha$ subunit of the ACDS from *Ca*. E. thermophilum are coordinated by residues similar to those described in the *M. barkeri* homologue, even if some substitutions in the close environments of the cofactors might tune their redox potentials (Supplementary Fig. 6). The C-cluster operating the CO-oxidation was obtained in a non-carbonylated state with the CO site vacant in both $\alpha$ subunits occupying the asymmetric unit (Fig. 2b). In contrast, the C-cluster in *M. barkeri* has been found to harbour a CO-bound ligand[17], explaining the differences in geometry in the overlay (Fig. 2b). Because of their gaseous and hydrophobic natures, the substrate CO and product $CO_2$ must transit in the enzyme through an internal hydrophobic channelling network that was experimentally identified in the structure of *M. barkeri*[17]. Computational analyses of the internal cavities confirmed the conservation of this channelling system in the $\alpha$ subunit of the complex from *Ca*. E. thermophilum (Fig. 2c and Supplementary Fig. 7). A main tunnel emanating from the C-cluster and ending up in two tunnels reaching the surface of the protein was detected and is proposed to be the CO-channel, as observed in the protein from *M. barkeri*[17] and the bacterial homologue *Clostridium autoethanogenum*[28] (Fig. 2c and Supplementary Fig. 7). An additional

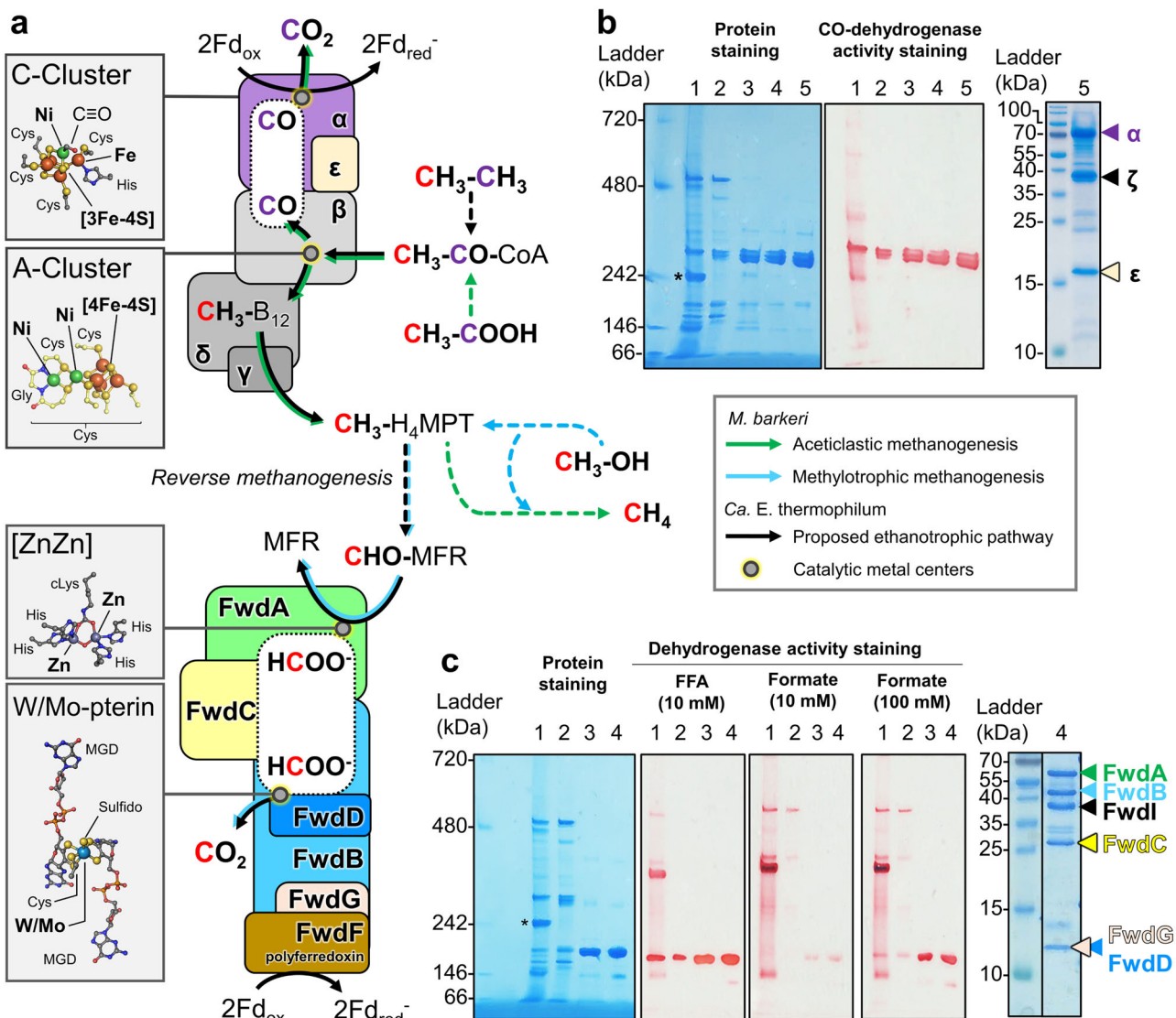

**Fig. 1 | Proposed catabolism of *Ca*. E. thermophilum, and native purification of the CODH component and the Fwd/Fmd complex. a** The pathway is based on studies described in *Methanosarcinales*[7,8,30]. The assembly of the ACDS (top) and Fwd (bottom) complexes are drawn in compliance with previous studies[17,19,62]. Arrows are coloured according to the corresponding metabolism, and dashed lines indicate multi-step transformations. White rounded rectangles framed by dotted lines illustrate internal channelling systems in which the substrates diffuse between both catalytic centres. Metallo-cofactor structures displayed in the inserts are derived from the deposited PDB models 1RU3 (acetyl-CoA synthase from *Carboxydothermus hydrogenoformans*[63]), 3CF4 (carbonylated ACDS α₂ε₂ subcomplex from *M. barkeri*[17]) and 5T5M (Fwd complex from *M. wolfei*[19]) with metals labelled in bold. cLys stands for carboxylysine. **b** Purification steps of the CODH component of the ACDS on native PAGE (left). (1) soluble extract; (2) anion exchange chromatography; (3, 4) hydrophobic exchange chromatography and (5) size-exclusion chromatography. The purified complex lacks the β, γ and δ subunits and, therefore, does not harbour the A-cluster and B₁₂. **c** Purification steps of the Fwd/Fmd complex on native PAGE (left). (1) soluble extract; (2) anion exchange chromatography; (3) hydrophobic exchange chromatography and (4) size-exclusion chromatography. **b**, **c** An asterisk marks the band corresponding to the ECR[11] on the native electrophoresis profile. **b**, **c** A denaturing (right) PAGE of the final enriched fractions. The electrophoresis profiles were similar for each purification. Source data are provided as a Source Data file.

internal cavity with ramifications was also predicted in the α subunits of the enzyme from *Ca*. E. thermophilum, yet a diffusion of the $CO_2$ and CO from the C-cluster to this extended intersubunit channelling system is improbable because a hydrophilic bottleneck restricts the gasses access to this cavity (Fig. 2c and Supplementary Fig. 7). The N-terminal extension of the α subunit, which cannot be modelled in the present structure, is in the vicinity of the end of the proposed CO tunnel (Supplementary Fig. 5). The extension is conserved in archaea, and is probably involved in the stabilisation of the other ACDS subunits (β, δ and/or γ), and especially the β subunit (acetyl-CoA decarbonylase) that should be connected to the CO tunnel (Fig. 1a). Based on the homology with the tunnelling system of *C. autoethanogenum* (PDB 6YTT,

Supplementary Fig. 7), the ACDS β subunit might exhibit a similar location in *Ca*. E. thermophilum.

The ζ subunit is positioned at the intersection of α and ε subunits (Fig. 2a, d and Supplementary Figs. 2, 3). It is composed of a ferredoxin domain in its N-terminal part (1–83, containing two [4Fe-4S] clusters), an $F_{420}$-reductase domain (84–349, containing one [4Fe-4S] cluster and the flavin adenine dinucleotide, FAD), and a C-terminal extension promoting the homodimeric interface (350–370). The $F_{420}$-reductase domain shares high structural similarity with the $F_{420}$-reducing subunit (FrhB) of the $F_{420}$-reducing hydrogenase from *Methanothermobacter marburgensis* (*Mm*Frh, PDB 4OMF[29]) and the $F_{420}H_2$-oxidising sulphite reductase Fsr (*Mj*FSR, PDB 7NP8[27]), with a comparable position and

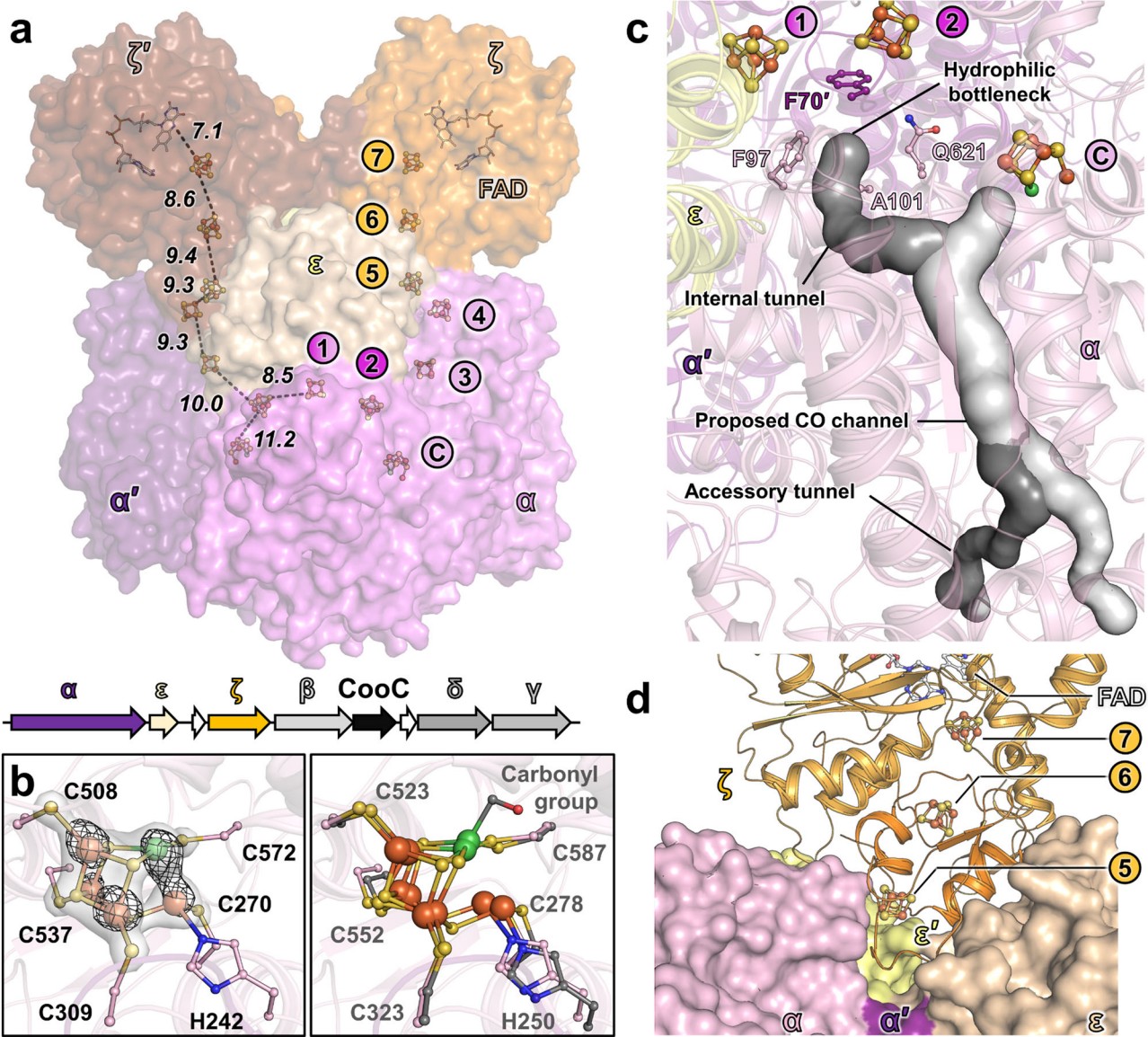

**Fig. 2 | CODH component of the ACDS from *Ca.* E. thermophilum. a** Overall structure of the subcomplex. The structure is shown as a surface. The distances between electron-transferring (metallo-)cofactors in the α′ε′ζ′ half-complex are presented by dashed black lines and given in Å. The genomic organisation of the genes encoding the complex is shown with labelled gene products and arrows coloured by subunits (white for unannotated, unrelated, or pseudogene) and size depending on the gene length. **b** Architecture of the C-cluster (left) and superposition with the homologue from *M. barkeri* (PDB 3CF4, coloured grey, right panel). The $2F_o − F_c$ and anomalous maps (collected at 12.67 keV), contoured at 3 and 5 σ, are shown as transparent white surface and black mesh, respectively. The carbonyl group modelled on the C-cluster of *M. barkeri* is absent in the structure of the ethanotroph. **c** The different tunnelling systems in α₂ε₂ζ₂ structure predicted by the CAVER programme are shown as surfaces and coloured by tunnels (structuring residues in Supplementary Fig. 7). **d** ζ dimer (cartoon) bound on the α₂ε₂ core (surface). For clarity, the protein chain after the ferredoxin-like N-terminal domain (residues 1 to 83) of the ζ subunit is coloured in light orange. The ζ′ subunit has been omitted for clarity. **a–d** Cofactors and residues are represented as balls and stick with oxygen, nitrogen, sulphur, phosphorus, iron and nickel coloured red, blue, yellow, light orange, orange and green, respectively. Carbons are coloured according to the respective chains and white for the FAD.

coordination of their (metallo-)cofactors (Supplementary Figs. 4, 8, 9 and Supplementary Table 4).

The anchoring of the ζ subunit to the α₂ε₂ core through its N-terminal domain (Fig. 2d) offers a structural template to picture how the electrons from the distal cluster of the α subunit (cluster 4) will be transferred on the distal cluster of the ζ subunit (cluster 5). The structure also depicts how a soluble ferredoxin would dock on ferredoxin-dependent archaeal ACDS. Compared to the bacterial system in which the D-cluster (homologous to cluster 1 in Fig. 2a) has been suggested to directly exchange electrons with the ferredoxin, the archaeal counterpart would exchange electrons via the peripheral cluster 4. This electron transfer is mandatory for $CO_2$-reduction during

acetyl-CoA synthesis and the reversal reaction during aceticlastic methanogenesis (Fig. 1a, green arrow). In the enzyme from *Ca.* E. thermophilum, the N-terminal domain of the ζ subunit, would act as an electron bridge to electronically connect the C-cluster to the FAD site of the ζ subunit (Fig. 2a), allowing the coupling of the CO-oxidation to $F_{420}$-reduction.

## Structure and composition of the formylmethanofuran dehydrogenase

The Fmd/Fwd complex was proposed to catalyse the second $CO_2$-generating step occurring in ethanotrophy[7,8]. These $CO_2$-releasing complexes have not yet been structurally unveiled, and their reaction

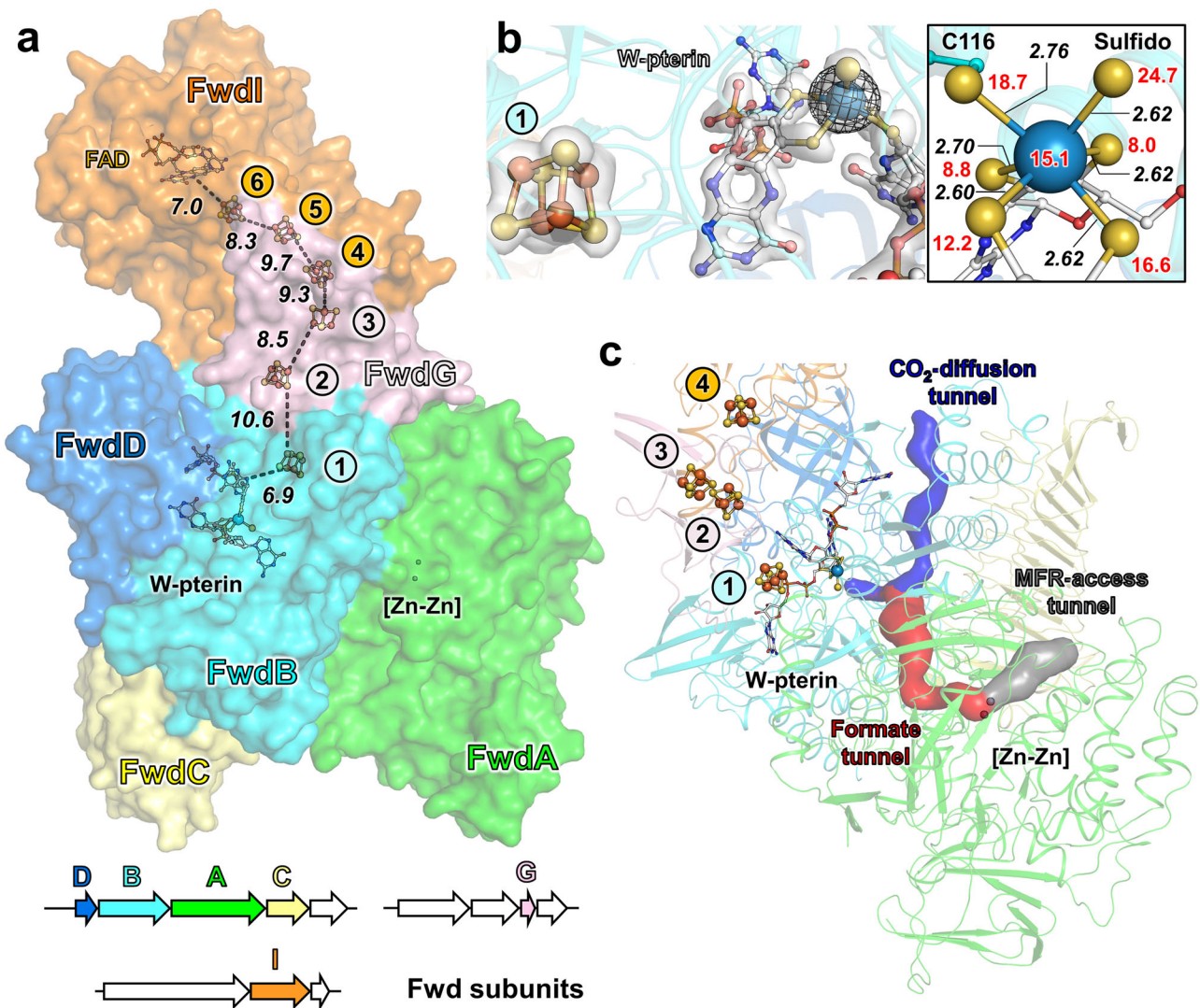

**Fig. 3 | Fwd complex from *Ca*. E. thermophilum. a** Overall structure of the Fwd complex. The proteins are represented as surfaces with the distance between electron-transferring cofactors highlighted by dashed black lines and given in Å. The genomic organisation of the genes encoding the complex is shown with labelled gene products and arrows coloured by subunits (white for unannotated, unrelated, or pseudogene) and size depending on the gene length. **b** Left, details of the tungstopterin site. The $2F_o − F_c$ (1.5 σ) and anomalous map (7.0 σ, collected at -12.40 keV) are shown as a white transparent surface and black mesh, respectively. Right, close-up of the tungsten ligands. Distances between the W atom and ligands are given in Å (black italic), and b-factors of atoms are given (red). **c** Tunnelling system in the Fwd complex from *Ca*. E. thermophilum. The tunnels predicted by the CAVER programme are represented as surfaces and coloured based on their proposed function (structuring residues in Supplementary Fig. 13). In all panels, the A, B, C, D, G, and I subunits are coloured green, cyan, light yellow, marine blue, light pink, and orange, respectively. Cofactors and residues are represented as balls and sticks with oxygen, nitrogen, sulphur, phosphorus, iron, zinc, and tungsten coloured red, blue, yellow, light orange, orange, light grey and blue-grey, respectively. Carbons from the cofactors are coloured white.

mechanisms were proposed to be analogous to the $CO_2$-fixing systems from hydrogenotrophic methanogens for which structures are available[19,20,22,30]. In this scenario, the binuclear [Zn-Zn] centre of the A subunit would hydrolyse the formyl-MFR into MFR and formate (or formic acid), this latter diffusing in an internal polar tunnel to the tungsto/molybdopterin site of the BD subunits to be oxidised into $CO_2$ with the concomitant reduction of ferredoxins (Fig. 1a).

The crystal structure of the Fmd/Fwd complex from *Ca*. E. thermophilum was refined at a resolution of 1.97 Å (Fig. 3a, Supplementary Table 2 and Supplementary Fig. 10a). Compared to the initial prediction of a molybdenum-containing enzyme[8], the anomalous data confirmed the presence of a tungsten atom in the active site, therefore renaming the complex Fwd (Fig. 3b).

The FwdABCDG core is similar to the structures of the $CO_2$-reducing formylmethanofuran dehydrogenase complex and sub-complex from *Methanothermobacter wolfei*[19] and *Methanospirillum*

*hungatei*[22] (Supplementary Fig. 10b). Both active sites, containing the metallocenters [Zn-Zn] and tungstopterin, are also structurally conserved (Supplementary Figs. 11, 12). In the latter, the tungsten is hexacoordinated by the Cys116, the two dithiolenes of the pterin ligand and a modelled sulfido ligand in a configuration similar to the high-resolution structure from *M. wolfei*[19]. Interestingly, the coordination of the pterin is almost identical in $CO_2$-fixing Fwd, suggesting that the preferentiality of the reaction lies in the electron acceptor/donor rather than fine-tuning of the active site environment. The internal tunnelling systems required for the transit of formyl-MFR, formate, and $CO_2$ can be predicted from the structure (Fig. 3c and Supplementary Fig. 13).

The Fwd complex forms a heterohexameric assembly in which the canonical polyferredoxin FwdF subunit is replaced by an $F_{420}$-reductase module (CAD7775209, named FwdI, Fig. 3a). As FwdF was reported to be required for large Fmd/Fwd-containing complexes or

supercomplexes[19,20,22], its substitution by the FwdI subunit prevents the formation of higher oligomeric states. Therefore, the lack of the polyferredoxin FwdF is coherent with the apparent absence of higher organisation in the soluble extract (Fig. 1c). The resulting Fwd complex is consequently the smallest ever structurally described. All subunits, with the exception of FwdG (probably because of its small molecular weight), were detected by mass spectrometry on the band obtained from native electrophoresis (Supplementary Table 3).

FwdI shares a similar organisation and (metallo)-cofactor content to the ζ subunit of ACDS, with the exception of the absence of the C-terminal extension involved in dimerisation (Supplementary Figs. 4b, 8, 9 and Supplementary Table 4). The six [4Fe-4S] clusters dispatched in FwdBGI would allow an efficient electron transfer between the tungstopterin to the FAD of FwdI (Fig. 3a), therefore possibly coupling formate oxidation to $F_{420}$-reduction.

### The CODH component of the ACDS and Fwd are all-in-one complexes reducing $F_{420}$ during substrate oxidation

The structural data gathered on the CODH component of the ACDS and the Fwd complex of *Ca*. E. thermophilum suggest that both enzymes catalyse $F_{420}$-reduction by the acquisition of a functional reductase (ACDS ζ subunit and FwdI, respectively). The access to the catalytic FAD in the ACDS ζ subunit and FwdI is surrounded by a positively charged surface that would stabilise the polyglutamate group of the $F_{420}$ tail, as proposed in other $F_{420}$-dependent enzymes (Supplementary Fig. 14). Our data show that the CODH component of ACDS and the Fwd complex from *Ca*. E. thermophilum can use $F_{420}$ as an electron acceptor for substrate oxidation, validating the functional assembly observed in the crystal structures (Fig. 4a, b). The measured rates of $F_{420}$ reduction in both complexes are lower than those of the methyl-viologen reduction. This difference has similarly been observed in the $F_{420}H_2$-oxidising sulphite reductase from *M. jannaschii*[31]. Here, it suggests that under these experimental conditions, the $F_{420}$-reductase module is the rate-limiting step compared to the substrate oxidation rates. In that case, methyl-viologen reduction rates will be higher because this artificial electron acceptor could directly uptake electrons near the substrate oxidative centre. It must also be noted that the $F_{420}$ used in this study was extracted from the *Methanococcales Methanothermococcus thermolithotrophicus*. Differences in the coenzyme structure compared to the native $F_{420}$ from *Ca*. E. thermophilum may limit the reaction kinetics. However, activity measurements with different $F_{420}$ concentrations do not suggest a low affinity for the coenzyme (Supplementary Fig. 15).

Both enzymatic complexes also accept ferredoxin, albeit with reduction rates eightfold lower than those for $F_{420}$ (Supplementary Fig. 15). We interpreted that the difference in activity is due to the presence of the additional $F_{420}$-reducing subunits that hinder the access to the electron transfer chain by the ferredoxin. Together, our experimental data argue that $F_{420}$ is the most relevant electron acceptor of both complexes under physiological conditions. It must be noted that alternative electron acceptors (e.g. flavodoxin) have already been described for purified bacterial ferredoxin-dependent CODH[32]. However, the CODH component from *Ca*. E. thermophilum is, to our knowledge, the first system depending on an additional reductase subunit to target a specific different electron acceptor. In comparison, methanogens, methanotrophs, and long-chain alkane-oxidising archaea[12–14] are supposed to strictly depend on ferredoxin for the reaction of ACDS and Fmd/Fwd complexes.

A comparison of the genomic environment of the genes coding for the ACDS and Fwd/Fmd complexes indicates that the coupling of $F_{420}$ reduction in ACDS and Fwd/Fmd complexes would generally be absent in methanogens and alkanotrophs (Supplementary Figs. 16, 17). Accordingly, the activities of $F_{420}$-reduction coupled to CO or furfurylformamide oxidation could not be detected in the cell extracts of enrichments of ANME-1 and ANME-2 or pure

cultures of the *Methanosarcinales M. barkeri* (Fig. 4c). The latter was also grown on acetate to illustrate the higher CO oxidase activity compared to methylotrophic methanogenesis (Fig. 1a, blue arrows), as ACDS becomes part of the catabolism during growth on acetate. However, even under growth on acetate, no $F_{420}$ reduction could be reliably measured. All gathered results point to an enzymatic coupling that seems to be a necessity for anaerobic ethane oxidation.

## Discussion

This study solved two crucial steps in the catabolic pathway of the ethane-oxidising archaeon *Ca*. E. thermophilum, deepening our knowledge of the ethanotrophy catabolism performed by the microbial community from deep-sea seeps. The illustrated native crystallisation approach, which could be applied to enrichment cultures of other slow-growing alkanotrophs, unveiled how these atypical archaea optimised their catabolism to derive cellular energy. The structural characterisation of these catabolic complexes illustrates once more how the evolution of microorganisms combines redox modules to cope with metabolic needs. The comparison of the $CO_2$-releasing CODH component of the ACDS complex and Fwd with $CO_2$-reducing homologues from other organisms shows a high conservation of the active sites. This suggests that metallo-cofactor modifications, coordination, or residue substitution in the active site would not dictate the reaction's directionality. It will be determined by metabolic fluxes or, in the present case, by the final electron acceptor of the reaction. This concept is particularly important when studying the catabolism of anaerobic alkanotrophs, for which the metabolic pathways still remain to be biochemically characterised.

The isolation of native systems from the thermophilic microbial enrichment revealed new pieces of the molecular puzzle of the anaerobic ethane oxidation, and the most intriguing is the key role of $F_{420}$ in this process. Relying on $F_{420}$ instead of ferredoxin as the final electron acceptor will affect the thermodynamics of the reactions. Considering the standard midpoint redox potentials of the CO-oxidation ($E°$ $CO/CO_2 = -520$ mV[33]) and formyl-MFR oxidation ($E°$ formyl-MFR/MFR + $CO_2 = -530$ mV[34]) coupled to $F_{420}$-reduction ($E°$ $F_{420}H_2/F_{420} = -340$ mV[24]), the $F_{420}$-reduction coupled to substrate oxidation catalysed by ACDS and Fwd would be highly exergonic. This could represent the thermodynamic pull of the anaerobic ethane oxidation, preventing the reversal of the pathway in the absence of ethane (i.e. $CO_2$ conversion to ethane by relying on $F_{420}H_2$-oxidation would be endergonic under physiological conditions). This might be particularly important in seeps, where concentrations of $CO_2$ largely exceed those of ethane. In the same line of thought, the exergonic process could counterbalance one or several unfavourable enzymatic reactions occurring during the uncharacterised conversion of ethyl-CoM into acetyl-CoA, explaining why the ζ and FwdI subunits are apparently conserved in the other cultured ethanotroph *Ca*. A. ethanivorans[7] (Supplementary Figs. 16, 17). Hence, this coupling would be specific to ethanotrophs as the metabolism of other alkanotrophs presents no selective pressure for such a thermodynamically favourable coupling. This agrees with the results presented, indicating that most methanogens and other alkanotrophs probably do not present such an enzymatic coupling (Fig. 4c and Supplementary Figs. 16, 17).

The $F_{420}$ reduction coupled with $CO_2$ generation also explains the peculiar energy conservation strategy of *Ca*. E. thermophilum, for which the genome does not encode ferredoxin-dependent cation pumps (i.e. Ech or Rnf complex). In contrast, the *Methanosarcina* genus uses the reduced ferredoxin pool derived from CO-oxidation or formyl-MFR oxidation to pump $H^+/Na^+$ by coupling ferredoxin oxidation to methanophenazine reduction or to generate $H_2$[30]. In the absence of the ferredoxin-dependent pumps, ferredoxin would not represent a viable electron carrier for energy conservation. Therefore, we hypothesise that this catabolism rather relies on the $F_{420}$ as a

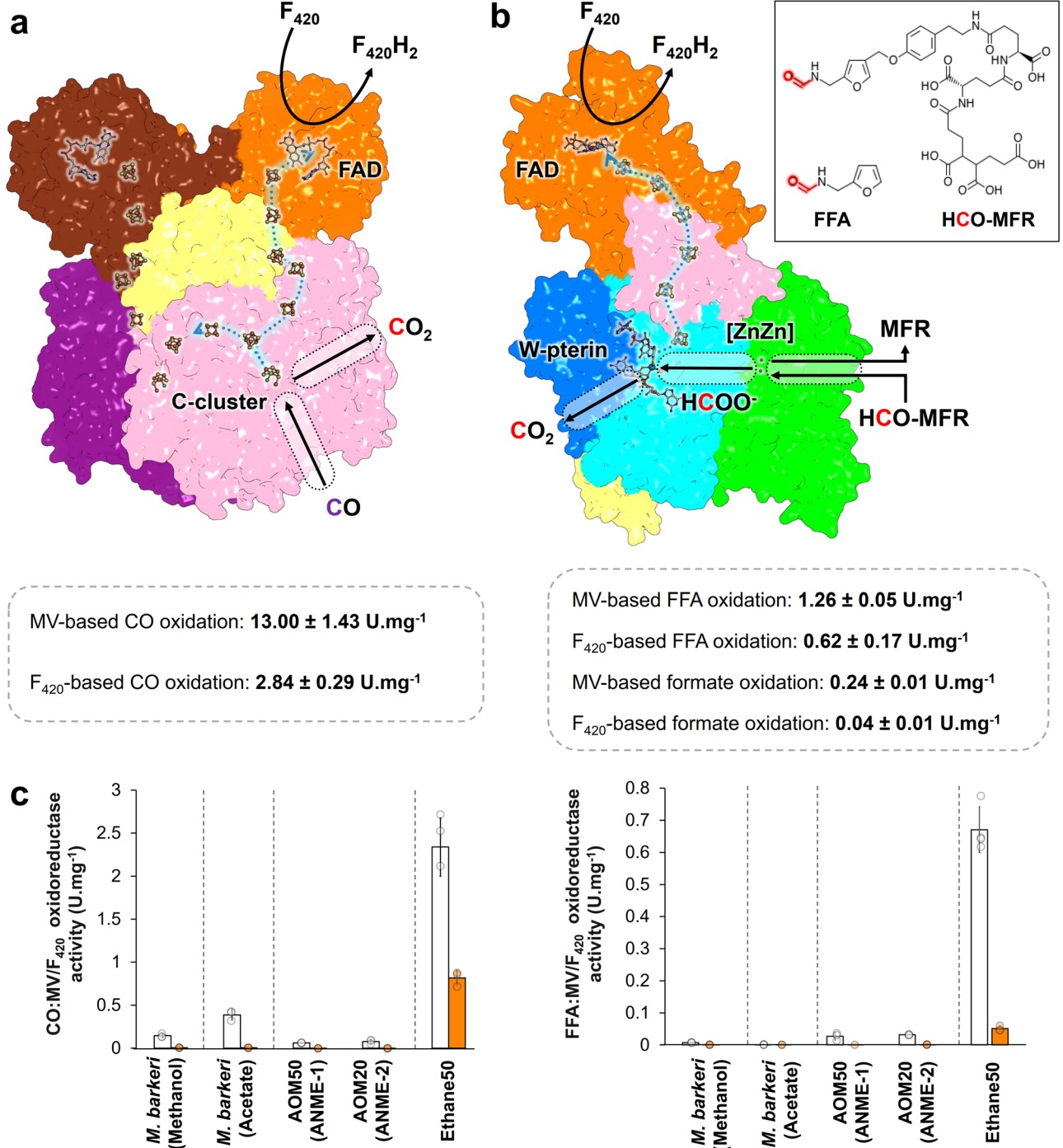

**Fig. 4 | Overall reactions and activity comparison of both complexes in cell extracts from ethanotroph, methanotrophs, and methanogens. a, b** Proposed reactions and activity measurements for CODH subcomplex (**a**) and Fwd complex (**b**). Water molecules and protons participating in the reactions have been omitted for clarity. Acetate (10 or 100 mM) was not used as a substrate by the Fwd complex with either MV or $F_{420}$. The chemical structure of formylmethanofuran (MFR) is the one described in *Methanothermobacter thermautotrophicus*[64]. **c** Activity measurements performed in soluble extracts from the ethanotroph, methanotrophs, and *M. barkeri* during aceticlastic and methylotrophic methanogenesis (the type of metabolism being separated by dashed lines). MV-based and $F_{420}$-based activities are coloured white and orange, respectively. Average and standard deviations are plotted, and the individual data are shown as transparent circles. **a–c** Activities are given in μmol of substrates oxidised (CO, FFA or formate) per minute per mg of pure enzyme or soluble proteins. Source data are provided as a Source Data file. Measurements were performed at least in triplicate ($n = 3$ or 4 independent measurements, see Source Data file) from a single extract for each organism or culture condition.

turntable electron carrier to drive ethanotrophy when dependent on a sulphate-reducing partner.

In the proposed metabolic model (Fig. 5), the highly expressed Fpo complex (Supplementary Table 5) is the only energy-conserving system that would allow ion translocations across the membrane to fuel the ATP synthase. The electrons flowing to the extracellular and membrane-bound quinones would be consumed by the thermodynamically favourable sulphate reduction pathway of the bacterial partner. The interspecies transfer would be operated through an elusive path that might imply conductive nanowires[8]. The stoichiometry

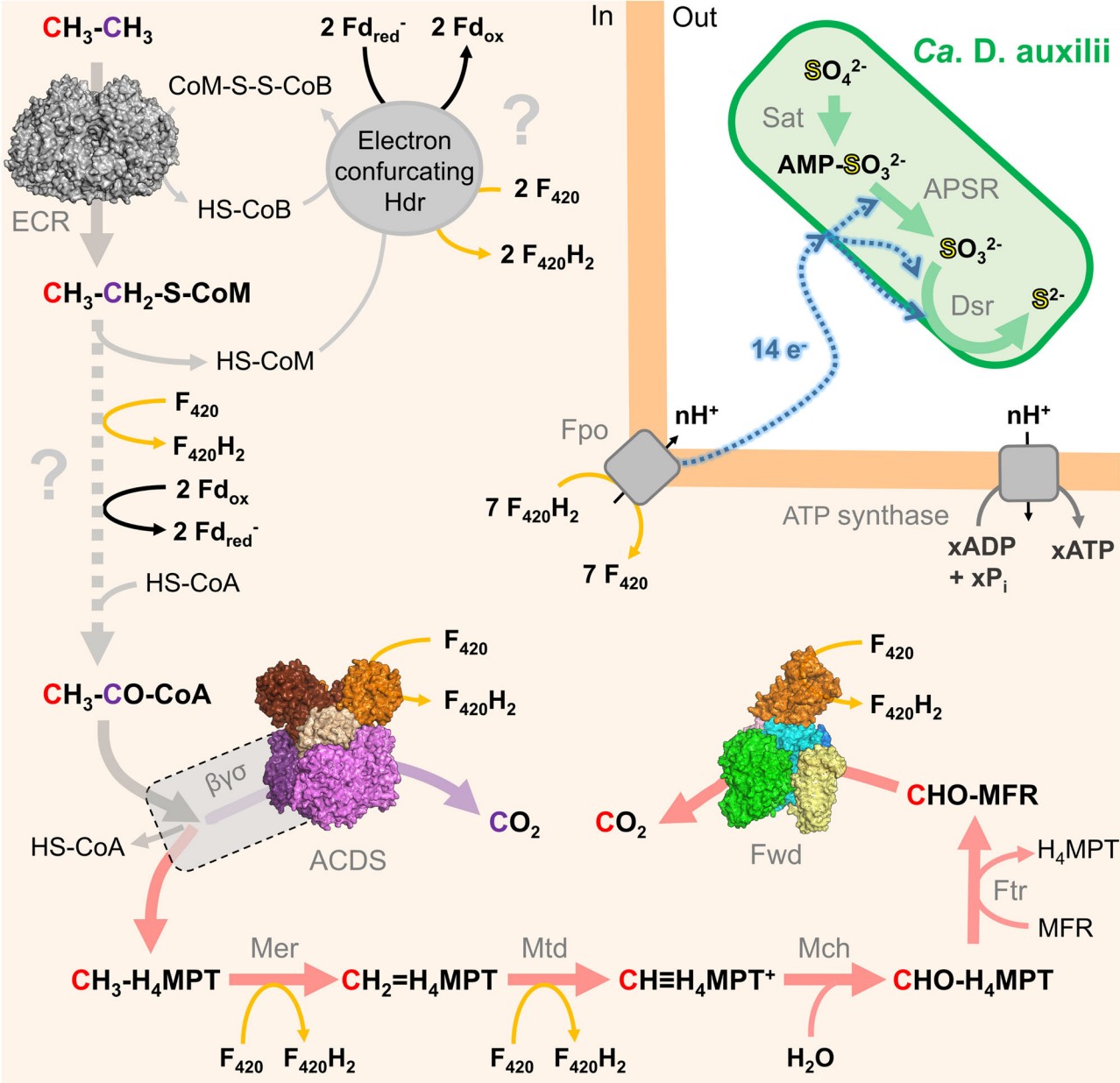

**Fig. 5 | Proposed catabolic metabolism in the Ethane50 consortium.** The structurally characterised enzymes are shown as surface representations, coloured as in Figs. 2, 3. The catabolic reactions are presented as large arrows coloured in grey, purple, red, and green, corresponding to the C2 part of ethanotrophy, ACDS activity, reverse methanogenesis, and sulphate reduction, respectively. A large dashed arrow indicates the yet uncharacterised ethyl-CoM to acetyl-CoA conversion. Orange arrows indicate $F_{420}$ reduction or $F_{420}H_2$ oxidation events. Question marks highlight the uncharacterised reactions that would employ ferredoxin. The ferredoxins are assumed to accept a single electron. The interspecies electron transfer is schematised in a blue dashed line, and the transfer mechanism was omitted in the figure for clarity. The exact number of ions translocated by the Fpo system is not known and is therefore labelled n and hypothesised to be protons. The ion/ATP ratio of the ATP synthase is also not known, and therefore, x ATP is produced, while ions are proposed to be protons. The stoichiometry of sulphate reduction is not respected for clarity.

of the ethane/sulphate oxidoreduction performed by the consortium (4 moles of ethane oxidised for 7 moles of sulphate reduced[8]) indicates that a total of seven $F_{420}H_2$ could be potentially obtained from the complete oxidation of one ethane molecule. The oxidation of acetyl-CoA by the ACDS and the reactions occurring in reverse methanogenesis (by the Fwd complex and the methylenetetrahydromethanopterin dehydrogenase and reductase) would reduce four out of the seven $F_{420}$. We propose that the missing reduced $F_{420}$s are derived from the two oxidative steps occurring during the metabolic transformation of ethyl-CoM to acetyl-CoA. Despite the uncharted nature of the metabolic pathway, we would expect a generation of one $F_{420}H_2$ and two reduced ferredoxins carrying one electron. The ferredoxins will be oxidised concomitantly with the heterodisulfide CoM-S-S-CoB, produced during ethyl-CoM generation, by the highly expressed putative $F_{420}$-reducing electron-confurcating heterodisulfide reductase to generate two $F_{420}H_2$ (Fig. 5 and Supplementary Table 5). The latter step is critical in the regeneration of the coenzyme employed for ethane capture. Since the carbon catabolic and anabolic pathways are expected to be separated, one would expect alternative $CO_2$-entry points or carbon sources for carbon assimilation. However,

the physiological utilisation of the $F_{420}H_2$ pool could be extended to assimilatory and anabolic pathways, as suggested by the numerous *frhB* homologues in the genomes of ethanotrophs (Supplementary Fig. 17). This unexplored reservoir of reactions coupled to $F_{420}(H_2)$ oxidoreduction must contain potential unknown metabolic routes and, among them, the reactions behind the ethyl-CoM transformation that remains to be elucidated.

## Methods

### Origin and cultivation of Ethane50, AOM50 and AOM20 enrichments

The Ethane50 enrichment used for enzyme purification derived from heated sediments of the Guaymas Basin hydrothermal vents. Sediments were sampled during the RV Atlantis mission AT 37-06 with submarine Alvin in December 2016. The culture conditions were previously described[8], and the studied biomass is similar to that used for the purification of the Ethyl-Coenzyme M Reductase (ECR), previously characterised[11]. An average of six to nine months is required to obtain sufficient biomass for one protein preparation. The AOM50 culture was derived from the Guaymas Basin (sampled during RV Atlantis mission AT 15-56 in December 2009). The culture conditions and microbial composition were described before[35–38]. The AOM20 culture, dominated by ANME-2 species, was derived from sediments from Amon Mud Volcano, Eastern Mediterranean Sea, retrieved during RV Atalante expedition Nautinil in 2003. Cultivation was performed at room temperature (20 °C) with methane as sole energy substrate. The microbial composition of this culture was described in ref. 39.

### Cultivation of *Methanosarcina barkeri*

*Methanosarcina barkeri* DSM 800 was obtained from the Deutsche Sammlung von Mikroorganismen und Zellkulturen GmbH, Braunschweig, Germany and was cultivated at 37 °C under strict anaerobic conditions in a medium whose composition was already described[40]. Sterile and anoxic methanol (1% v/v) or 100 mM sodium acetate were added as carbon and energy sources. The initial gas phase contained $N_2/CO_2$ (90:10%) at 50 kPa. The overpressure coming from the metabolic activity during growth was regularly evacuated. The cells were harvested in the late-exponential phase by centrifugation for 30 min at $17,000 \times g$ and kept frozen at −80 °C under anaerobic conditions before use.

### Protein extraction and purification

Cells collected from exponentially growing Ethane50 culture were used for protein extraction. The medium was removed with a stainless steel needle by applying an overpressure of $N_2/CO_2$ (with a 90:10% ratio). After 3-min-long flushing with $N_2/CO_2$ (90:10%), the cells were pelleted by centrifugation for 15 min at $16,250 \times g$ in an anaerobic chamber filled with an $N_2/CO_2$ atmosphere (90:10%) at room temperature, and the supernatant was removed. Cells were suspended in SR medium[41] and stored at −80 °C under an $N_2/CO_2$ (90:10%) atmosphere until purification. The presence of black aggregate precipitates hindered the precise quantification of the available biological material.

Because of the extremely limited biomass, only three purification procedures (representing between 55 to 100 mg of total soluble protein) have been performed and were used for this work. Purifications were performed with a similar purification protocol. Cell lysis and preparation of extracts were performed in an anaerobic chamber filled with an $N_2/CO_2$ atmosphere (90:10%) at room temperature. A volume of 15 ml of sedimented cells was suspended in 16 ml of 50 mM tricine/NaOH buffer pH 8.0 and 2 mM dithiothreitol (DTT; buffer A). The lysis protocol included a sonication step (BANDELIN Sonopuls HD 2200) followed by five rounds of French press at around 1000 PSI (6.895 MPa), yielding a homogenous deep-black extract. The French press cell was flushed with $N_2$ and washed twice with anoxic buffer A before use. The soluble extract was obtained by ultracentrifugation at

140,000×g for 1 h at 4 °C. Enzyme purification was carried out under anaerobic conditions in a Coy tent filled with an $N_2/H_2$ atmosphere (97:3%) at 20 °C and under yellow light. For each step, chromatography columns were washed with at least three-column volumes (CV) with the corresponding loading buffer, and samples were filtrated on 0.2 μm filters prior to loading. During purification, the enzymes were followed by high-resolution clear native polyacrylamide gel electrophoresis (hrCN PAGE, see below), sodium dodecyl sulphate PAGE (SDS PAGE), and absorbance monitoring at 280, 415 and 550 nm.

Extracts were diluted with buffer A to obtain a final 15-fold dilution before being loaded on $4 \times 5$ ml anion exchanger HiTrap™ Q HP columns (GE Healthcare) equilibrated with the same buffer. After a 2 CV washing, proteins were eluted with a 0 to 0.40 M NaCl linear gradient for 6 CV at a 1 ml.min$^{-1}$ flow rate. The fractions containing both the CO dehydrogenase (CODH) and the formylmethanofuran dehydrogenase (Fwd) complexes from *Ca*. E. thermophilum eluted between 0.38 M and 0.40 M NaCl. The pooled fractions were diluted with 2 volumes of an anoxic 50 mM Tris/HCl buffer pH 7.6 and 2 mM DTT (buffer B) containing 2 M ammonium sulphate before being loaded on a Source™ 15PHE 4.6/100 PE (GE Healthcare) equilibrated with the same buffer. After washing, proteins were eluted with a 1.60 to 0 M ammonium sulphate linear gradient for 53 CV at a 1 ml.min$^{-1}$ flow rate.

The CODH component eluted between 1.13 and 0.80 M ammonium sulphate. In one of the purifications, the hydrophobic exchange chromatography step using the Source™ 15PHE 4.6/100 PE column has been performed twice to enhance sample purity (see Fig. 1b and Supplementary Table 1). Fractions of interest were pooled, concentrated on a 30-kDa cut-off centrifugal concentrator (nitrocellulose, Vivaspin from Sartorius) and optionally injected on a Superdex 200 Increase 10/300 GL. The size-exclusion chromatography was performed in 25 mM Tris/HCl buffer pH 7.6, 10% (v/v) glycerol, and 2 mM DTT (buffer C) at a 0.4 ml.min$^{-1}$ flow rate. The CODH complex eluted in a Gaussian peak.

The Fwd complex eluted between 1.50 M and 1.13 M ammonium sulphate during hydrophobic exchange chromatography. Fractions of interest were pooled and concentrated on a 30-kDa cut-off centrifugal concentrator (nitrocellulose, Vivaspin from Sartorius). Optionally, the concentrated pool was injected on a Superdex 200 Increase 10/300 GL. The size-exclusion chromatography was performed in buffer C at a 0.4 ml.min$^{-1}$ flow rate. The Fwd complex eluted in a Gaussian peak.

Both purified proteins were directly used for anaerobic crystallisation and enzymatic activities or stored at −80 °C in anoxic conditions in the case of the measurements shown in Supplementary Fig. 15.

Protein concentration was estimated by the Bradford method (Bio-Rad Laboratories, Munich, Germany) for all samples. A bovine serum albumin (BSA) standard was used to estimate protein concentration.

### Preparation of cell extracts

The different extracts used for activity measurements presented in Fig. 4c were prepared in an anaerobic chamber filled with an $N_2/CO_2$ atmosphere (90:10%) at room temperature. The extracts from *M. barkeri* were obtained from anaerobically frozen pellets of 4.24 and 1.15 g (wet weight) of cells grown on methanol and acetate, respectively. The biomass used for the preparation of the extracts from ANMEs was obtained from the sedimentation of the cultures of AOM20 and AOM50 (around 1.5 and 2 ml sedimented cells, respectively). For all organisms, the biomass was diluted with 5 ml of buffer B before a sonication step (BANDELIN Sonopuls HD 2200). Five rounds of French Press at around 1000 PSI (6.895 MPa) were used as an additional lysis step. Unbroken cells and debris were removed by ultracentrifugation at 140,000×g for 1 h at 4 °C. Protein concentration was estimated by the Bradford method using BSA as standard. The protein concentration from the ANMEs extracts was increased by using a 10-kDa cut-off centrifugal concentrator (nitrocellulose, Vivaspin from Sartorius).

## High-resolution clear native polyacrylamide gel electrophoresis (hrCN PAGE)

The hrCN PAGE protocol was adapted from ref. 42. The electrophoresis was performed in an anaerobic chamber filled with a $N_2/CO_2$ (90:10%) atmosphere. Fresh anaerobic samples were used. Glycerol (20% v/v final) was added to samples, and 0.001% (w/v) Ponceau S was used as a protein migration marker. The anaerobic electrophoresis cathode buffer contained a buffer mixture of 50 mM tricine/NaOH, 15 mM Bis-Tris at a pH 7.0 supplemented with 0.05% (w/v) sodium deoxycholate, 0.01% (w/v) dodecyl maltoside and 2 mM DTT. The anaerobic anode buffer contained 50 mM Bis-Tris buffer, pH 7.0, and 2 mM DTT. hrCN PAGEs were carried out using an 8 to 15% linear polyacrylamide gradient, incubated overnight in an anaerobic tent, and soaked in an anaerobic cathode buffer. Gels were run with a constant 20 mA current using a PowerPac™ Basic Power Supply (Bio-Rad). After electrophoresis, gels were stained with Instant Blue™ (Expedeon) or used for enzymatic staining (see below). Estimation of complex size on gels was performed by measuring the distance between the centre of the sample wells and the centre of the protein bands in each lane. The three different bands formed by the CODH subcomplex were treated separately in the lanes in which they could be clearly identified. The protein ladder (NativeMark unstained protein ladder, Fischer Scientific) was used to establish the standard curve. The equations derived from the standard curves were used to estimate the size of the standards and both enzymatic complexes.

## Peptide identification by liquid chromatography–tandem mass spectrometry

Protein bands produced by native electrophoresis were subjected to mass spectrometry for peptide identification. The bands of interest were cut out and washed multiple times with 150 μl of destaining buffer (25 mM ammonium bicarbonate, 50% (v/v) ethanol) and then dehydrated with 150 μl of 100% ethanol. After suspending the ethanol, the gel pieces were dried using vacuum centrifugation. Next, 50 μl of digestion buffer (25 mM Tris/HCl, 10% acetonitrile (v/v), 10 ng/μl trypsin) was added. The mixture was incubated on ice for 20 min, followed by the addition of 50 μl of ammonium bicarbonate buffer (25 mM), and the gel pieces were incubated overnight at 37 °C. Peptides in the supernatant were collected, and additional peptides were extracted from the gel pieces through repeated incubation at 25 °C in 100 μl of extraction buffer (3% (v/v) TFA, 30% (v/v) acetonitrile), followed by centrifugation and collection of the supernatants. Finally, the gel pieces were dehydrated by incubating at 25 °C in 100 μl of 100% acetonitrile, and the supernatant was combined with the supernatants from the previous extraction steps. Acetonitrile was removed by vacuum centrifugation, and 70 μl of 2 M Tris/HCl, 10 mM tris(2-carboxyethyl)phosphine (TCEP), and 40 mM 2-chloroacetamide (CAA) were added. After a 30-min incubation at 37 °C, the peptides were acidified to 1% (v/v) TFA and loaded onto Evotips (Evosep).

Evotips were eluted onto a 15-cm PepSep C18 column (15 cm × 15 cm, 1.5 μm, Bruker Daltonics) using the Evosep One system. The column was maintained at 50 °C, and peptides were separated employing the 30 sample per day (SPD) method. The eluting peptides were directly sprayed onto the Orbitrap Exploris 480 mass spectrometer (Thermo Fisher). For data-dependent acquisition (DDA), the full mass scan range was set from 300 to 1650 m/z with a resolution of 60,000. Up to 15 of the top precursors were selected for fragmentation using higher energy collisional dissociation (HCD) with a normalised collision energy of 28. MS2 spectra were recorded at a resolution of 15,000. The AGC targets for MS and MS2 scans were set at 300 and 100%, respectively, with maximum injection times of 25 ms for MS scans and 28 ms for MS2 scans. Dynamic exclusion was set to 30 ms.

Raw data were processed using the MaxQuant computational platform[43] (version 2.0.10) with standard settings applied. Shortly, the peak list was searched against a database composed of the protein sequences predicted from the genome assembly of the sequences of *Ca*. D. auxilii (ASM157752v1) and *Ca*. E. thermophilum (GoM-Arc1_E50) with an allowed precursor mass deviation of 4.5 ppm and an allowed fragment mass deviation of 20 ppm. Cysteine carbamidomethylation, methionine oxidation and N-terminal acetylation were set as variable modifications.

The raw data were deposited on the public PRIDE database (EMBL-EBI) under the following deposition number: PXD054507.

## Determination of the molecular weight and oligomeric state by size-exclusion chromatography

The chromatography was performed on a Superdex 200 Increase 10/300 GL (GE Healthcare, Munich, Germany) in 25 mM Tris/HCl pH 7.6, 2 mM DTT, and 10% (v/v) glycerol at a 0.4 ml.min$^{-1}$ flow rate and in an anaerobic Coy tent containing an $N_2/H_2$ (97:3%) atmosphere. The elution volumes from a high molecular weight range gel filtration calibration kit (GE Healthcare, Munich, Germany) were used to establish the protein standard curve. Estimations of the molecular weight from the standards and the CODH component and Fwd complex were derived from the equations of the standard curves.

## Enzymatic assays

The enzymatic activity measurements performed during purification and on purified enzymes were performed in 50 mM Tris/HCl pH 7.6 and 2 mM DTT under anaerobic conditions and at 50 °C. A final concentration of 5 mM methyl-viologen (MV) was used as an electron acceptor, its reduction being spectrophotometrically followed at 600 nm on a Cary 60 UV-Vis Spectrophotometer (Agilent Technologies) in sealed quartz cuvettes prepared under an $N_2/CO_2$ (90:10%) atmosphere. A molar absorption coefficient of 14,181 M$^{-1}$.cm$^{-1}$ was experimentally determined under these conditions and used for calculations. The presented activities are in μmol of the oxidised substrate (CO, furfurylformamide or formate) per min per mg of protein, considering that 2 moles of MV are reduced per mole of the oxidised substrate. For CODH activity, the reaction was initiated by the addition of 0.2 ml of 100% CO (99.997% purity, Air Liquide) in an anaerobic quartz cuvette of 1.4 ml containing 1 ml of the reaction mixture. For furfurylformamide/formate dehydrogenase activity measurements, activity was started by the addition of furfurylformamide (95% purity, ChemSpace, 10 mM final) or formate (10 or 100 mM final) solutions prepared in anoxic deionised water. The protein concentration ranged from 17.57 μg.ml$^{-1}$ for the soluble extract to 0.41 μg.ml$^{-1}$ for the purified fractions. No Fwd activity could be detected with either 10 or 100 mM sodium acetate.

The $F_{420}$ reduction by both enzymes was monitored in similar conditions, using 18.8 μM of $F_{420}$ purified from *M. thermolithotrophicus*. $F_{420}$ was prepared by following the previously published protocol[27]. If the molecule was shown to be a suitable electron acceptor for the complexes, the endogenous $F_{420}$ from the ethanotroph could be different, affecting the activity of the complexes. $F_{420}$ reduction was monitored at 420 nm using an experimentally determined molar absorption coefficient of 41,503 M$^{-1}$.cm$^{-1}$. The presented activities are in μmol of oxidised substrate per min per mg of protein, considering that one mole of $F_{420}$ is reduced per mole of the oxidised substrate. About 50 μM of flavin adenine dinucleotide (FAD) and flavin mononucleotide (FMN) was added to the enzyme stock before activity measurement to warrant flavin saturation and maximal activity of the $F_{420}$-reducing subunits. The final concentration of FAD and FMN during activity measurement was inferior or equal to 0.1 μM. The addition of FAD and FMN did not affect measurements in the absence of the enzymes. The protein concentration ranged from 0.16 to 5 μg.ml$^{-1}$. Activity in soluble extracts was monitored in similar conditions in 0.4 or 1 ml reaction volume, except that FAD and FMN concentration was 100 μM. The final concentration of FAD and FMN during activity measurement was inferior or equal to 1.2 μM. The addition of FAD and

FMN did not affect measurements in the absence of the extracts. Experiments were run at least in triplicates, the number being clearly shown as individual data points and available in the Source Data file. Activities measured in the extracts of *M. barkeri*, AOM50, AOM20, and Ethane50 were performed at 37, 50, 20 and 50 °C, respectively. The protein concentration ranged from 4.4 to 62.4 µg.ml$^{-1}$. The indicated units refer to µmole of CO or furfurylformamide oxidised per minute per mg of protein, using the parameters and coefficients listed above.

The activities shown in Supplementary Fig. 15 were performed similarly on enzymes stored frozen at −80 °C. FAD and FMN were not supplemented with pure proteins. The activity of the CO dehydrogenase was assayed in 0.6 ml quartz cuvettes containing 0.2 ml of reaction mixture. The furfurylformamide dehydrogenase activity was assayed in quartz cuvettes or in microplates containing 0.2 ml of reaction mixture and monitored by a BMG Labtech FLUOstar Omega Microplate reader in an anaerobic chamber filled with an N$_2$ (100%) atmosphere. The ferredoxin used was directly extracted from the acetogenic bacterium *C. autoethanogenum*[44]. The ferredoxin reduction was monitored at 420 nm, with an experimentally determined molar extinction coefficient of 14,154.8 M$^{-1}$.cm$^{-1}$ used for calculation and assuming a one-electron reduction of ferredoxin. Therefore, we considered that 2 moles of ferredoxin are reduced per mole of the oxidised substrate for activity calculation reported in Supplementary Fig. 15. The enzyme concentration ranged from 8.6 to 13.86 µg.ml$^{-1}$.

In-gel viologen-based activity staining was performed as previously described[28]. Shortly, activity staining was performed in 10 ml of anoxic 50 mM Tris/HCl buffer pH 7.6, 2 mM DTT, 5 mM MV and 1 mM of 2,3,5-triphenyltetrazolium chloride (TTZ). The latter component was used for oxygen-insensitive staining. Native gels were loaded with 10, 5 or 2 µg of extracts or pure enzymes. After electrophoresis, gels were transferred to a 0.5 l Duran bottle containing the staining solution described above. The gas phase was changed for 100% N$_2$ at 50 kPa. For the CODH activity, 20 ml of 100% CO was added. Formate (10 or 100 mM final) or furfurylformamide (10 mM final) was added for Fwd activity. The reaction was started by the addition of 1 ml of an anoxic solution of 50 mM MV and 10 mM TTZ (final 5 mM MV and 1 mM TTZ). Reactions were performed at 50 °C and stopped by opening the bottle under a fume hood and transferring the gel to aerobic deionised water.

The equations of the monitored reactions are (FA and FFA stand for furfurylamine and furfurylformamide, respectively):

**MV – based CO oxidation** : $CO + H_2O + 2\,MV_{ox} \rightarrow CO_2 + 2H^+ + 2MV_{red}^-$

**F$_{420}$ – based CO oxidation** : $CO + H_2O + F_{420} \rightarrow CO_2 + F_{420}H_2$

**MV – based FFA oxidation** : $FFA + H_2O + 2\,MV_{ox} \rightarrow CO_2 + 2H^+ + FA + 2MV_{red}^-$

**F$_{420}$ – based FFA oxidation** : $FFA + H_2O + F_{420} \rightarrow CO_2 + FA + F_{420}H_2$

**MV – based formate oxidation** : $HCOO^- + 2\,MV_{ox} \rightarrow CO_2 + H^+ + 2MV_{red}^-$

**F$_{420}$ – based formate oxidation** : $HCOO^- + H^+ + F_{420} \rightarrow CO_2 + F_{420}H_2$

## Protein crystallisation

All crystals were obtained at 20 °C in a Coy tent under an N$_2$/H$_2$ (97:4%) by using the sitting drop method on 96-Well MRC 2-Drop Crystallisation Plates in polystyrene (SWISSCI). The crystallisation reservoir contained 90 µl of mother liquor. Crystallisation drops contained a mixture of 0.6 µl protein and 0.6 µl precipitant.

Crystals of the α$_2$ε$_2$ζ$_2$ ACDS subcomplex from *Ca*. E. thermophilum were obtained in 20% (w/v) polyethylene glycol 6000, 100 mM MES pH 6.0, and 200 mM ammonium chloride (JBS Wizard™ crystallisation screen from Jena Bioscience). The initial protein concentration was 13.0 mg.ml$^{-1}$, and 1 mM final of both FAD and FMN was added to the protein before crystallisation.

Crystals of the Fwd complex from *Ca*. E. thermophilum were obtained in 18% (w/v) polyethylene glycol 8000, 200 mM sodium acetate trihydrate and 100 mM sodium cacodylate, pH 6.5 (SG1™

crystallisation screen from Molecular Dimensions). The initial protein concentration was 4.64 mg.ml$^{-1}$.

## X-ray data collection, model building and refinement

Crystals of the α$_2$ε$_2$ζ$_2$ ACDS subcomplex and the Fwd complex were soaked for a few seconds in the crystallisation condition supplemented with 30% (v/v) glycerol before being frozen in liquid nitrogen under anoxic conditions prior to X-ray diffraction studies. All diffraction experiments were performed at 100 K. The structure of the α$_2$ε$_2$ζ$_2$ ACDS subcomplex was initially solved by performing a single anomalous dispersion experiment at the Fe K-edge (Supplementary Table 2) using the *SHELX* package[45] (CCP4 package version 8.0.004), using diffraction data collected on the PXIII beamline (X06DA) from the Swiss Light Source (SLS, Villigen, Switzerland). Due to the relatively low resolution, these data were only used to experimentally solve the substructure and initiate the tracing of the secondary structure. The resolution was extended to 1.893 Å by a dataset collected at a wavelength of 0.98 Å on the PROXIMA-1 beamline from SOLEIL (Paris-Saclay, France).

The data for the α$_2$ε$_2$ζ$_2$-ACDS subcomplex and the Fwd complex were processed and scaled with *autoPROC*[46] (version 1.0.5). The data were accordingly further processed with *STARANISO* correction integrated with the *autoPROC* pipeline[47]. Crystallographic data presented anisotropy along the following axes: a = 1.892 Å, b = 2.155 Å and c = 2.301 Å for the α$_2$ε$_2$ζ$_2$-ACDS subcomplex and a = 1.885 Å, b = 2.291 Å and c = 3.018 Å for the Fwd complex.

The structure of the Fwd complex was solved by molecular replacement using the Fwd complex from *M. thermolithotrophicus* (PDB 5T5M[19]) as a template, using diffraction data collected on the PXIII beamline (X06DA) from the SLS.

All models were manually built with *COOT*[48] (version 0.9.8.3 EL) and refined with *PHENIX* (version 1.20.1-4487). The last refinement steps were performed by refining with a translation libration screw (TLS) and were validated by the MolProbity server[49] (http://molprobity.biochem.duke.edu). Both models were refined with hydrogens in the riding position. Hydrogens were omitted in the final deposited models. The PDB ID codes of the structures are 8RIU and 8RJA for the α$_2$ε$_2$ζ$_2$-ACDS subcomplex and the Fwd complex, respectively. Data collection and refinement statistics for the deposited models are listed in Supplementary Table 2. Analysis of the b-factors of the cofactors indicates a 100% occupancy in both models. A portion of the electronic density and omit maps of the cofactors for both structures are shown in Supplementary Figs. 18, 19.

## Structural analyses

All figures were generated and rendered with PyMOL (Version 2.2.0, Schrödinger, LLC, New York, NY, USA). Internal tunnel predictions were performed by the *CAVER* tool[50]. The analysis of the α$_2$ε$_2$ζ$_2$ ACDS subcomplex was performed by applying a probe radius of 1.0 Å and starting from the Ni atom of the C-cluster. The analysis of the Fwd tunnelling system was performed by applying a probe radius of 1.0 Å and starting from the sulfido ligand of the tungstopterin, ignoring the Zn atoms of the [Zn-Zn] binuclear centre.

## Bioinformatic analyses

The proteins used for model construction and phylogenetic analysis were extracted from the genome assemblies from *Ca*. E. thermophilum (GenBank: LR991654.1[8]), *Ca*. A. ethanivorans (GenBank: RPGO01000021.1[7]), *M. barkeri* MS (Genbank CP009528.1), *Methanothermobacter wolfei* isolate AS20ysBPTH_75 (GenBank JAAYMZ010000098.1[51]), ANME-1 isolates (GenBank JAGGRR010000017.1[52], JAFNKG010000061.1[53], QENH01000201.1[54], PQXC01000010.1 and PQXB01000001.1), *Candidatus* Methanoperedens nitroreducens (GenBank JAIOIS010000036.1[55]), *Candidatus* Syntrophoarchaeum butanivorans (GenBank LYOR01000001.1[13]),

*Candidatus* Syntrophoarchaeum caldarius (GenBank. LYOS01000001.1[13]), *Candidatus* Methanoliparum thermophilum (GenBank RXIF01000004.1[56]) and *Candidatus* Methanoliparum whitmanii, *Candidatus* Methanoliparum widdelii and *Candidatus* Methanoliparum zhangii (obtained from the NODE database under the Data name/data ID/Analysis ID: XY_O_T55_M2_bin.61/OED248975/ OEZ00007026, HX_O_T65_bin.11/ OED00802402/OEZ00007017 and GD_Cm_T35_P3_bin.32/OED00802400/OEZ00007012, respectively[14], available at https://www.biosino.org/node/analysis/detail/[Analysis ID]). The protein sequences and putative operon organisations were obtained from the genomes using the Operon Mapper webserver[57]. The genomes used for analysis, as well as the outputs from the analysis from Operon Mapper, are deposited on Zenodo under the following https://doi.org/10.5281/zenodo.13381200[58]. Gene length was taken into consideration when creating figures. The protein annotation of ACDS and Fwd/Fmd subunits are derived from the available structures or by a BLAST search against the predicted protein sequences using as query the sequences from the different subunits of the characterised ACDS from *M. barkeri*[17], Fwd complex from *M. wolfei*[19] and $F_{420}$-reducing subunit of the $F_{420}$-reducing hydrogenase[59]. The percentages of identity and query coverages presented are extracted from the BLAST analysis. The ferredoxin-like FwdG subunit could not be used as a query because of the similarity with too many irrelevant proteins. The BLAST search results using the chaperon CooC as a query that was not in the genomic area encoding for other ACDS subunits were not conserved. Proteins with less than 50% coverage and/or 30% sequence identity were considered irrelevant. Only continuous alignments were conserved in the analysis. Due to the modular nature of polyferredoxins, proteins exhibited a size too different from the FwdF query (with a cut-off of around 10%) were not selected for the analysis. Pterin-dependent enzymes, functionally composed by a fusion of FwdB and FwdD, were neither considered as FwdB nor FwdD homologues. No coverage or sequence identity could have been derived from the interrupted sequences (e.g. a gene coding for an isoform of the ACDS α subunit in *M. wolfei*). The other proteins were manually annotated by BLAST research in the PDB and SwissProt databanks. The phylogenetic tree was constructed with sequences of all proteins homologous to ACDS ζ subunit and FwdI using the maximum likelihood method and was generated with MEGA X (version 10.1.8)[60] by using an alignment constructed with *MUSCLE*[61]. A total of 200 replicates were used to calculate node scores. Functions are derived from the available literature or by similarity with existing enzymes.

## Statistics and reproducibility

The microbial distribution of the ethane-oxidising enrichment is assumed to be relatively stable over time of cultivation. The different purifications did not suggest any significant change in the microbial population of protein abundancy.

The presented results for activity measurements were performed separately using the same procedure and the same extract or pure enzyme. The individual data, average and standard deviation are presented. The presented activity rates and electrophoresis profiles are representative of distinct purifications performed from different cultures of the same organisms or enrichment (except for extracts from ANME species, whose cultivation time and yield did not allow to reproduce the experiment).

X-ray diffraction data collected from distinct protein crystals yielded similar enzyme structures, the highest resolution being kept for modelling and analysis. Crystallisation was however only attempted once due the drastic protein limitation.

## Reporting summary

Further information on research design is available in the Nature Portfolio Reporting Summary linked to this article.

## Data availability

All structures were validated and deposited in the Protein Data Bank (PDB) under the following accession numbers: 8RIU, Crystal structure of the $F_{420}$-reducing carbon monoxide dehydrogenase component [https://www.rcsb.org/structure/8RIU] and 8RJA, Crystal structure of the $F_{420}$-reducing formylmethanofuran dehydrogenase complex [https://www.rcsb.org/structure/8RJA]. All other data were available in the manuscript or the supplementary materials. Source data are provided in this work. The mass spectrometry raw data generated in this study have been deposited in the PRIDE database (EMBL-EBI) under accession code PXD054507. The genomic sequences and the outputs of the Operon Mapper server generated in this study have been deposited in the Zenodo public database[58] under accession code [https://zenodo.org/records/13381200]. Source data are provided with this paper.

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

## Acknowledgements

We are deeply thankful to Cedric J. Hahn for his help in the cultivation and participation in the enzyme purification. We thank the Max Planck Institute for Marine Microbiology and the Max Planck Society for their continuous support. We thank the SOLEIL and Swiss Light Source (SLS) synchrotrons for beam time allocation and the respective beamline staffs of PROXIMA-1 and X06DA for assistance with data collection. We also acknowledge Christina Probian, Ramona Appel, and Mélissa Belhamri for their invaluable support in the Microbial Metabolism research group. We thank the Mass Spectrometry Core Facility at the Max Planck Institute of Biochemistry for the peptide identification. Additional funds came from the Deutsche Forschungsgemeinschaft (DFG) funding the Cluster of Excellence The Ocean Floor—Earth's Uncharted Interface (EXC-2077–390741603) at MARUM, University Bremen and the DFG project ETHOX (WA 4053/2-1 and WE 5492/1-1). The initial crystallisation screening performed by an OryxNano robot was supported by the DFG priority programme 1927 Iron-Sulphur for Life WA 4053/1-1.

## Author contributions

O.N.L., G.W. and T.W. designed the research. O.N.L. and G.W. performed cultivation and culture experiments. O.N.L. and T.W. purified and crystallised the proteins, collected X-ray data, built the models, analysed the structures, interpreted the data and wrote the paper, with contributions and final approval of all co-authors. O.N.L. performed the activity measurements.

## Funding

## Competing interests

The authors declare no competing interests.
