## [Peer Review file · Nature Communications]

Ethane-oxidising archaea couple CO₂ generation to F₄₂₀ reduction

Corresponding Author: Dr Tristan Wagner

Version 0:

Reviewer comments:

Reviewer #1

(Remarks to the Author)

General comments:

It is interesting that complexes of the F₄₂₀ reductase with both the CO dehydrogenase and the formylmethanofuran-dehydrogenase can be isolated from this ethane-oxidizing organism. However, there is insufficient evidence supporting the overarching statement that the organism reorients its overall metabolism towards F₄₂₀ reduction. In addition, the manuscript does not report the structure of a so-called "ACDS/F₄₂₀ reductase complex", but instead of a CODH/F₄₂₀-reductase. Acetyl-CoA decarbonylase/synthase (ACDS) was not even assayed or shown to be present in their complex. It is quite a stretch to state that the manuscript describes "the biochemical, structural and enzymatic characterization of both complexes". The results are rather incremental in only slightly advanced our understanding of alkane metabolism.

Finally, this manuscript appears to have been hurriedly prepared and was a difficult read. For example, in terms of organization, after Extended Data 2, a bunch of Figures appear, including repeats of the main data figures – and some that have strange formatting and lack Figure legends. Furthermore, the poor grammatical structure throughout makes the paper ambiguous and often misleading, though I'm sure this ambiguity was not intentional. I've made some comments and suggested edits. I hope that the authors seek guidance in extensively revising the manuscript for eventual publication elsewhere, especially with respect to the placement of clauses (which should modify the noun directly proceeding them) and the subject/verb agreement.

Detailed Comments:

The Science:

1. The ACS does not appear to be part of the complex, so it is incorrect to state that they have isolated ACDS.
2. Fig. 1. It isn't clear from Fig. 1 what complexes that the authors isolated. And the text isn't so explicit either. The SDS gel indicates that there were three proteins in the complex: CODH (alpha/epsilon) and the 420 reductase. They should explicitly use the term zeta on the band for the zeta subunit of the F₄₂₀ reductase on the SDS gel, since that term is so extensively used in the paper.
3. In Fig. 1d, they showed cartoons of the zeta 2 dimer (cartoon) "bound on the alpha2epsilon2 core (surface)". But, based on the SDS gel, it doesn't appear that the "ferredoxin-like N-terminal" is part of their complex. This figure might be OK for a review. But it is very misleading and confusing in relating the findings of this paper. The term "as a non-transparent cartoon" is also odd verbage.
4. I did not see any mention of the fractional occupancy of the metal centers. Biochemical characterization of the complexes is incomplete. How much cobalamin, what state is the cobalamin in? How active is the complex? Does the complex catalyze acetyl-CoA synthesis? CO/acetyl-CoA exchange?
5. There are no numbers on the many supplementary (Extended Data) Figures.
6. There are little biochemical data in the Main Figure section of the paper to support their conclusions and is limited to some activity measurements in Fig. 4 and Extended Data Table 1. The first column of "activity" should be labeled as Total Units.
7. In Extended Data 2, the number of atoms presented should include a percentage completeness.
8. After Extended Data 2, a bunch of Figures appear, including repeats of the main data figures – and some that have strange formatting and with no Figure legends. Presumably these are to be ignored??
9. Results, Line 66: "CO-oxidation activity carried by the CO-dehydrogenase subunit (CODH, α subunit)" – Isn't the epsilon subunit required for CODH activity? Has the methanogenic alpha subunit ever been isolated alone?
10. What is the ratio of the subunits based on densitometry scans of the SDS gel bands (relative to the molecular mass of the

subunits).

11. Related to the title and throughout the paper, there is insufficient evidence supporting the concept that this ethanotroph reorients its overall metabolism towards F420 reduction", because they performed CO- and formylmethanofuran-dependent F420 reduction capabilities of the two complexes and compared this to methyl viologen. Does they reduce ferredoxin? The key parameter related to use of F420 as a substrate is the k_{cat}/K_m for F420, compared to that with other electron acceptors (like ferredoxin, etc.) It is a large extrapolation to generalize this limited kinetic analysis even to making conclusions about the individual enzymes and an even larger leap to extend this to overall cellular metabolism.

12. Line 81: "ACDS ... decompose in several populations on native PAGE, probably due to electrophoresis-dependent artefacts coming from a partial loss of cofactors or stabilising ions". It is more likely due to the low affinity of the component subunits.

13. Regarding the structures, the Redundancies, reflections, CC1/2s: (0.999 and 0.989 (0.621), and $I/\sigma I$ [10.8 and 7.2) values are good, and Rvalues seem fine, indicating no over-fitting.

14. Line 59: "questioning the dogma of the CO₂-releasing step coupled to ferredoxin in these organisms." I question that the statement that CODH-ferredoxin is dogma. I'm sure that the authors can find some papers that study Fd as electron acceptor for CODH, however, a number of papers, starting from the earliest ones, point out how promiscuous CODH is in its electron carrier selectivity. Furthermore, CODH has been shown to reduce both flavodoxin and flavins, so it is not surprising that it also can reduce F420, a deazaflavin. Perhaps this should be cited within the paper.

Composition issues:

1. Abstract: "methanogenic or methanotrophic relative". Please clarify the meaning of this. It seems unlikely that the CODH and formate dehydrogenases in ethanotrophic organisms are so different in their ability to reduce F420, that the methanotrophic and methanogenic enzymes are unable to reduce F420. It would be very interesting if this is true and unambiguously demonstrated.

2. Abstract: "coupling C1-oxidation". Be specific about what compounds are being oxidized here, formate and CO.

3. Intro: "Only two archaeal species belonging to the Methanosarcinales order can catalyse the complete anaerobic oxidation of ethane". This is ambiguous. Do the authors mean that there are only two archaeal species (no others exist on earth) that catalyze ethane oxidation? Or that only two species have been isolated so far that catalyze ethane oxidation and they both are Methanosarcina?

4. Intro (line 46-7): "Therefore, ethane would be ultimately converted into two molecules of CO₂ and CO₂-releasing steps (ACDS and Fwd/Fmd complexes) would be coupled to ferredoxin reduction," As written, this also is ambiguous/incorrect, because grammatically it indicates that "CO₂-releasing steps (ACDS and Fwd/Fmd complexes) would be coupled to ferredoxin reduction". Please clarify. It would probably be more correct to state that "Therefore, ethane oxidation (involving/or including ACDS and Fwd/Fmd complexes) to CO₂ would be coupled to ferredoxin reduction".

Reviewer #2

(Remarks to the Author)

The manuscript by Lemaire et al, utilises native purification and x-ray crystallography to structurally characterise the CODH and Fwd complexes from the *Achaea Candidatus Ethanoperedens thermophilum*, grown in an enrichment culture. *Achaea Candidatus Ethanoperedens thermophilum* is one of the few microbes capable of the anaerobic oxidation of ethane, which it performs syntrophically with a sulphate reducing bacteria. The CODH and Fwd complexes are thought to play key a role in this process, through the oxidation of carbon monoxide and formyl-MFR.

The manuscript is well written and comprehensively describes the detailed structural characterisation of these complexes, which is backed up by functional and bioinformatic analysis. The authors approach to obtaining these complexes is innovative, and the challenge of natively isolating these complexes from anaerobic enrichment cultures cannot be overstated. The authors should be commended for their technical expertise in producing a sample that is high quality enough to generate crystals that diffract to high resolution.

The key finding of the study is that both the CODH and Fwd complexes contain a module homologous to a F420-reductase, strongly implying that these complexes donate electrons from CO and formyl-MF to the redox cofactor F420. The authors support this with activity assays of the purified complexes and cell lysates showing that CODH, Fwd and cell lysates from the *Ethane50* enrichment culture can use F420 as an electron acceptor, while cell lysates from ANME-1/2 and *M. barkeri* cannot.

The authors then propose a model based on these data where *Achaea Candidatus Ethanoperedens thermophilum* use F420 as their main electron acceptor, which has the advantage of making ethane oxidation exergonic, creating favourable thermodynamic conditions for its oxidation.

Overall, the study is well conducted, and the structural data is compelling. However, I think it is necessary that the authors address the relatively low reaction rates observed for CODH, Fwd and *Ethane50*, when F420 is utilised as the electron acceptor. For the purified enzymes these rates are significantly lower than the non-specific electron acceptor methyl viologen. It isn't intuitive to me why this would be the case if F420 is the physiological electron acceptor substrate for the enzymes.

Obviously, it would be great to obtain a crystal structure of CODH and Fwd in complex with F420 or a derivative, but I realise

this may not be a reasonable request given the amount of data already in the study. However, molecular docking and/or structural analysis (given the negatively charged glutamate tail of F420, there is likely to be a corresponding positively charged region at the acceptor substrate binding site) to support F420 binding would be appropriate. Additionally, further activity measurements in the presence additional acceptor substrates could further clarify the physiological electron acceptor for these enzymes.

Version 1:

Reviewer comments:

Reviewer #1

(Remarks to the Author)

The manuscript by Lemaire et al, describes predominantly x-ray crystallographic analysis to reveal the structure of the CODH and Fwd complexes from the *Achaea Candidatus Ethanoperedens thermophilum*. This is one of the few microbes capable of the anaerobic ethane oxidation, which it performs syntrophically with a sulphate reducing bacteria. The CODH and Fwd complexes play key a role in ethane oxidation, through the oxidation of carbon monoxide and formyl-MFR. The manuscript comprehensively describes the detailed structural characterisation of these complexes, which is backed up by some functional and bioinformatic analysis. The authors should be commended for meeting the challenge of natively isolating these complexes from anaerobic enrichment cultures and producing a sample that is high quality enough to generate crystals that diffract to high resolution. The revision is improved. There are just a few additional changes in toning down some of the hyperbolic language in favor of perhaps more accurate language. Otherwise, I think it's a useful contribution to the literature.

1. The authors continue to state that they : "broke the long-standing dogma" that CO₂-forming reactions in the acetyl-CoA reducing pathways are coupled to ferredoxin.) No. It has been demonstrated that CODH is promiscuous in its redox partner choice. What has been known since CODH was first isolated is that this enzyme has a very wide range of electron acceptors. The actual electron acceptor in the case of F420 appears to be a 4Fe-4S metal center in the F420 reductase. They need to compare Fd and F420 as electron acceptors in terms of CO oxidation to validate the point that the CODH from ethane-oxidizing organisms have a special redox property ("Accordingly, both systems exhibit robust coupled F420-reductase activities, which are not detected in the cell extract of related methanogens and anaerobic methane oxidisers." Furthermore, is incorrectly stated that CODH-dependent reduction of flavodoxin and rubredoxin is a "side reaction". The cited study looked at various redox partners including ferredoxin, flavodoxin and rubredoxin alone. They all accept electrons from CODH.

It should be made clear that the actual electron acceptor for CODH is an FeS cluster in the F420 reductase, which reduces the F420.

2. The manuscript should include color coding on each figure to identify the proteins and subunits.

3. While they have partly clarified the nature of the protein (CODH, not ACDS), they continue to state that they performed an enzymatic characterization (instead of performing an enzyme assay). In terms of nomenclature, there are still places where they talk about ACDS instead of CODH. For example, it is not ACDS that is CO₂ releasing and especially in their complex, it is CODH.

Reviewer #2

(Remarks to the Author)

The authors have responded in detail to my comments/suggestions regarding the manuscript, and implemented appropriate changes to the manuscript to address them. This is a very nice and well rounded study. I don't have any further comments or concerns

Response to the reviewer's comments

Reviewer #1 (Remarks to the Author):

General comments:

It is interesting that complexes of the F₄₂₀ reductase with both the CO dehydrogenase and the formylmethanofuran-dehydrogenase can be isolated from this ethane-oxidizing organism. However, there is insufficient evidence supporting the overarching statement that the organism reorients its overall metabolism towards F₄₂₀ reduction. In addition, the manuscript does not report the structure of a so-called "ACDS/F₄₂₀ reductase complex", but instead of a CODH/F₄₂₀-reductase. Acetyl-CoA decarbonylase/synthase (ACDS) was not even assayed or shown to be present in their complex. It is quite a stretch to state that the manuscript describes "the biochemical, structural and enzymatic characterization of both complexes". The results are rather incremental in only slightly advanced our understanding of alkane metabolism.

We thank the reviewer for the time invested in the reviewing process and the interest in our work. The different raised points are discussed in detail in the below sub-sections, as well as the answers to the different remarks and suggestions with the modifications of the manuscript being underlined.

- 1) *"insufficient evidence supporting the overarching statement that the organism reorients its overall metabolism towards F₄₂₀ reduction"*.

Due to the analogy to the methanogenesis pathway and for thermodynamic reasons, the CO₂-releasing reactions catalysed by the ACDS and Fwd complexes were previously proposed to reduce ferredoxin (Fd, Chen *et al.*, 2019, Hahn *et al.*, 2020). However, such an Fd-dependent core metabolism cannot occur in *Ca. E. thermophilum* because of the absence of genes coding for Fd-dependent energy-conserving membranous complexes. Therefore, an alternative energy-conserving system must operate by relying on a different central electron carrier, as mentioned in the revised introduction.

Lines 48-56. "Therefore, ethane would be ultimately oxidised into two molecules of CO₂, and the CO₂-releasing enzymes (ACDS and Fwd/Fmd complexes) are expected to reduce ferredoxin, which is employed for energy conservation in methanogens¹⁵. The electrons released during ethane oxidation are supposed to be indirectly or physically transferred to sulphate-reducing bacteria living in a syntrophic partnership with the archaea^{7,8}." The ethanotrophs do not contain any known membranous systems that would allow energy conservation from ferredoxin oxidation, questioning if the CO₂-releasing step operated by ACDS and Fwd/Fmd would be necessarily coupled to ferredoxin reduction, an established dogma in methanogens and alkanotrophs."

The reorganisation of the metabolism towards F₄₂₀ reduction was a working hypothesis to construct a coherent metabolic model of the ethane-oxidizing consortium. It was based on the previous genomic and transcriptomic data collected on the organism (Hahn *et al.*, 2020) and studies on the interaction of microorganisms in alkane oxidising archaea (i.e., Wegener and Krukenberg *et al.*, 2015, Krukenberg *et al.*, 2018). The hypothesis of ethane-oxidising catabolism depending on F₄₂₀ is supported by our experimental data gathered on the CODH subcomplex and Fwd complex and bioinformatics analyses on the genome. However, it is correct that the reactions catalysing the transformation of ethyl-CoM to acetyl-CoA have not been demonstrated. Our claim that such reactions are expected to deliver one reduced F₄₂₀ and two reduced Fd (with one electron carried by ferredoxin) to allow the heterodisulfide production by electron confurcation are speculative.

The text was modified to stress the hypothetical nature of this model and to tune down the conclusions regarding the overall metabolism.

Lines 23-27: "Based on these crystal structures, enzymatic activities, and metagenome mining, we propose a model in which the catabolic oxidising steps orchestrated by the ethanotroph would wire electron delivery to the F₄₂₀. Via this specific adaptation, the electron transfer from reduced F₄₂₀ to the sulphate-reducing partner would fuel energy conservation and represent the driving force of ethanotrophy."

Lines 64-67: "The biochemical, structural and enzymatic characterisation of both complexes supports that the CO₂-generating steps are coupled to F₄₂₀ reduction instead of ferredoxin, suggesting that F₄₂₀ reduction is the main driver of this metabolism²⁴."

Lines 291-293: “In the proposed metabolic model (Fig. 5), the highly expressed Fpo complex (Supplementary Table 3) is the only energy-conserving system that would allow ions translocation across the membrane to fuel the ATP synthase.”

- 2) *“the manuscript does not report the structure of a so-called “ACDS/F420 reductase complex”, but instead of a CODH/F420-reductase. Acetyl-CoA decarbonylase/synthase (ACDS) was not even assayed or shown to be present in their complex”*

The CO-oxidizing enzyme described here is indeed not a complete ACDS complex but rather a partial subcomponent obtained after destabilisation of the ACDS, probably during cell lysis. The phrasing may have been unclear in the previous version, but we tried our best to clarify the case. Firstly, we confirmed the CODH subunit composition by additional mass spectrometry data, confirming that the only peptides present are the subcomplex catalysing CO-oxidation and F₄₂₀-reduction. Therefore, other activities (e.g. acetyl-CoA decarbonylation) could not be measured. Secondly, we rewrote the text accordingly. Please see below for a detailed explanation.

- 3) *“It is quite a stretch to state that the manuscript describes “the biochemical, structural and enzymatic characterization of both complexes”.”*

We recognise that our work does not have a complete enzymatic characterisation (e.g., kinetic parameters have not been determined). However, we would like to underline the exceptional nature and technical challenge of this study. These enzymes have been purified from the native organism, which cannot be isolated so far due to symbiotic requirements. The slow-growing enrichment culture and its intrinsic microbial diversity (i.e., only 30-40 % of the biomass corresponds to the organism of interest) is a hurdle to derive sufficient biomass and isolate the protein complexes from the highly heterogeneous mixture. One year of cultivation yields a microbial pellet from which 55-100 mg of the total amount of soluble proteins could be purified (most biochemical studies on native organisms combining structural studies start with 1,000 - 5,000 mg of total protein from a monoclonal microbial population). Yet, the experimental data provided here is comparable to that previously published for other archaeal ACDS and formyl-MFR dehydrogenases. Therefore, achieving the purification, structural characterisation, and specific enzymatic activities of catabolic enzymes from this mixture is a tour de force that has been acknowledged in our previous study (Hahn *et al.* 2021).

Because of this biomass limitation, we could unfortunately not determine all biochemical and enzymatic properties. However, we sincerely believe that the specific enzymatic activities (comprising our newly presented data) and the structural analyses decipher the key points: the subunit and (metallo)cofactor composition and electron acceptor identity.

A sentence has been added to stress that point: Lines 350-351: “Because of the extremely limited biomass, only three purification procedures (representing between 55 to 100 mg of total soluble protein)”.

- 4) *“The results are rather incremental in only slightly advanced our understanding of alkane metabolism.”*

To our knowledge, this study is the first enzymatic characterisation of native enzymes from an alkanotrophic archaeon (as the reported MCR from ANME-1 purified from Black Sea mats did not contain activities and stressed the importance of working on a native system). By showing that formyl-MFR dehydrogenase and CO dehydrogenase are coupled to an F₄₂₀ reductase subunit via experimental validation, we broke the long-standing dogma that CO₂-forming reactions in the acetyl-CoA reducing pathways are coupled to ferredoxin.

For these reasons, we would instead argue that the presented results are crucial in understanding the anaerobic oxidation of ethane. However, we agree with the referee that the presented specific metabolic strategy could be different in other alkanotrophs.

This work highlights the unique nature of the ethanotrophic pathway by providing a structural knowledge extendable to the enzymatic C1-chemistry. The presented formyl-MFR dehydrogenase is only the third structure described as an enzyme from this superfamily and the first of its kind described in the *Methanosarcinales* order. More excitingly, this is the first structure of an Fwd performing CO₂ generation as a physiological reaction, as all other structurally characterized homologues are known to fix CO₂. We also describe the second structure of a partial ACDS complex from archaea, providing a picture of the ferredoxin-binding site in archaeal homologues.

Finally, this manuscript appears to have been hurriedly prepared and was a difficult read. For example, in terms of organization, after Extended Data 2, a bunch of Figures appear, including repeats of the main data figures – and some that have strange formatting and lack Figure legends. Furthermore, the poor grammatical structure throughout makes the paper ambiguous and often misleading, though I'm sure this ambiguity was not intentional. I've made some comments and suggested edits. I hope that the authors seek guidance in extensively revising the manuscript for eventual publication elsewhere, especially with respect to the placement of clauses (which should modify the noun directly preceding them) and the subject/verb agreement.

We are sorry that the manuscript was difficult to read and apologise if the format caused any discomfort to the reviewer. The misleading repetition of figures after the Extended Data Table 2 comes from the addition of each individual high-resolution figure, as suggested in the submission process. This caused the “merged PDF” to contain additional information but ensured the figures were of high quality (e.g., the PDF conversion could have decreased the resolution of the embedded figures).

We could have taken the risk of resubmitting without providing the high-resolution figures in TIFF format. However, we still added them based on a common agreement so all reviewers could inspect the original figures before possible deterioration during the PDF conversion. Therefore, we would politely ask this reviewer to ignore the additional figures following the Extended Data Table 2 in the revised merged PDF file.

The manuscript has undergone major modifications by simplifying the sentence structures as suggested.

Detailed Comments:

The Science:

1. The ACS does not appear to be part of the complex, so it is incorrect to state that they have isolated ACDS.

The purified CO-oxidising enzyme is indeed a subcomplex of the ACDS super-complex. The isolated enzyme contains α , ϵ and ζ subunits. That is why we initially labelled the purified $\alpha_2\epsilon_2\zeta_2$ enzyme as “ACDS subcomplex”. We are confident that the purified $\alpha_2\epsilon_2\zeta_2$ enzyme belongs to the ACDS complex for the following reasons: (1) the genome of *Ca. E. thermophilum* encodes a single isoform of α and ϵ subunits. These genes are embedded in the operon harbouring the other subunits of the ACDS complex (see Figure 2a). (2) The genes coding for the ACDS complex are highly expressed, supporting the catabolic role of the ACDS for ethane oxidation (Hahn *et al.*, 2020, Supplementary Table 3). (3) Such a partial ACDS complex was also previously obtained from *Methanosarcina barkeri* (Gong *et al.*, 2008), probably due to complex destabilisation during purification or after cell lysis. Accordingly, the native electrophoresis indicates that the $\alpha_2\epsilon_2\zeta_2$ component of the ACDS complex is already separated from the rest of the ACDS complex in the soluble extract. We performed additional mass spectrometry experiment to validate the presence of α , ϵ and ζ subunits (Supplementary Table S1) and the absence of the other expected peptides (β , γ and δ subunits).

To clarify that point, we referred to this purified enzyme as the “CODH component of the ACDS”, a nomenclature previously used for *M. barkeri* (Gong *et al.*, 2008).

Lines 56-60: “To solve this metabolic puzzle, we here characterise the multi-enzymatic ACDS and Fwd/Fmd complexes¹⁶⁻²³ by isolating the CODH component of the ACDS and the entire Fwd complex directly from a microbial enrichment of a syntrophic consortium composed of the ethane-oxidising archaeon *Candidatus Ethanoperedens thermophilum* and the sulphate-reducing bacterium *Candidatus Desulfoservidus auxilii*.”

Lines 69-70: “The isolated CODH component of ACDS and the Fwd/Fmd complex differ from characterised homologues”

Lines 104-113: “All three subunits were detected by mass spectrometry analysis from the band exhibiting CODH activity on native PAGE, while no peptide belonging to the three other ACDS subunits (β , δ and γ) could be detected in this sample (Supplementary Table 1). The purified CODH is supposed to be a subcomponent of the ACDS for the following reasons: (i) the α and ϵ subunits are the sole isoforms encoded in the *Ca. E. thermophilum* genome, (ii) the genes encoding for the CODH subunits are in the same genomic region encoding the other ACDS subunits (Fig. 2a), and (iii) genes coding for the ACDS complex are expressed in these culture conditions, and the enzyme is supposed to be part of the ethanotrophy pathway^{7,8}. We suppose that the whole ACDS complex was destabilised upon cell lysis (e.g., due to the low affinity of the component subunits), simplifying the characterisation of the CODH component.”

Line 208-210: “The structural data gathered on the CODH component of the ACDS and Fwd complexes of *Ca. E. thermophilum* suggests that both enzymes catalyse F₄₂₀-reduction by the acquisition of a functional reductase (CODH ζ subunit and FwdI, respectively).”

Lines 213-215: “Our data shows that the CODH component of the ACDS and the Fwd complex from *Ca. E. thermophilum* can use F₄₂₀ as an electron acceptor for substrate oxidation, validating the functional assembly observed in the crystal structures (Fig. 4a and b).”

Lines 255-257: “. The comparison of the CO₂-releasing CODH component of the ACDS complex and Fwd complexes with CO₂-reducing homologues from other organisms shows a high conservation of the active sites.”

Lines 875-876: “b, Purification steps of the CODH component of the ACDS on native PAGE (left).”

2. Fig. 1. It isn't clear from Fig. 1 what complexes that the authors isolated. And the text isn't so explicit either. The SDS gel indicates that there were three proteins in the complex: CODH (alpha/epsilon) and the 420 reductase.

We stressed and clarified that point in the text. In addition, we added the mass spectrometry data (as Supplementary Table 1) confirming by peptide identification the subunits composing the purified CODH component of ACDS and Fwd complex.

Lines 104-113: “All three subunits were detected by mass spectrometry analysis from the band exhibiting CODH activity on native PAGE, while no peptide belonging to the three other ACDS subunits (β, δ and γ) could be detected in this sample (Supplementary Table 1). The purified CODH is supposed to be a subcomponent of the ACDS for the following reasons: (i) the α and ε subunits are the sole isoforms encoded in the *Ca. E. thermophilum* genome, (ii) the genes encoding for the CODH subunits are in the same genomic region encoding the other ACDS subunits (Fig. 2a), and (iii) genes coding for the ACDS complex are expressed in these culture conditions, and the enzyme is supposed to be part of the ethanotrophy pathway^{7,8}. We suppose that the whole ACDS complex was destabilised upon cell lysis (e.g., due to the low affinity of the component subunits), simplifying the characterisation of the CODH component.”

Lines 114-116: “The α₂ε₂ core of the CODH component can be reliably superposed on the homologous structure from *Methanosarcina barkeri*¹⁷, also obtained from dissociation from the ACDS complex (Extended Data Fig. 1a and b).”

They should explicitly use the term zeta on the band for the zeta subunit of the F420 reductase on the SDS gel, since that term is so extensively used in the paper.

As proposed, the subunits were labelled with their respective names on the figure for clarification (ζ and FwdI).

3. In Fig. 1d, they showed cartoons of the zeta 2 dimer (cartoon) “bound on the alpha2epsilon2 core (surface)”. But, based on the SDS gel, it doesn't appear that the “ferredoxin-like N-terminal” is part of their complex. This figure might be OK for a review. But it is very misleading and confusing in relating the findings of this paper. The term “as a non-transparent cartoon” is also odd verbage.

The “ferredoxin-like N-terminal” referred to in Figure 2d is not a distinct peptidic chain but the N-terminal domain of the ζ subunit. It refers to the detailed subunit description in the text:

Lines 149-153: “The ζ subunit is positioned at the intersection of α and ε subunits (Fig. 2a and 2d, Supplementary Fig. 2). It is composed of a ferredoxin domain in its N-terminal part (1-83, containing two [4Fe-4S] clusters), an F₄₂₀-reductase domain (84-349, containing one [4Fe-4S] cluster and the flavin adenine dinucleotide, FAD), and a C-terminal extension promoting the homodimeric interface (350-370).”

Two shades of orange have replaced the transparent cartoon in Figure 2d. The legend of Figure 2d was modified for clarification:

Lines 895-897: “d, ζ₂ dimer (cartoon) bound on the α₂ε₂ core (surface). For clarity, the protein chain after the ferredoxin-like N-terminal domain of the ζ subunit (formed by residues 1 to 83) of the ζ subunit is coloured in light orange.”

4. I did not see any mention of the fractional occupancy of the metal centers. Biochemical characterization of the complexes is incomplete. How much cobalamin, what state is the cobalamin in? How active is the complex? Does the complex catalyze acetyl-CoA synthesis? CO/acetyl-CoA exchange?

As explained previously, the presented CODH component (composed of α , ϵ and ζ subunits) is a portion of the ACDS complex. This can be seen from the SDS and native PAGE profile, the obtained structure and the additional mass spectrometry data presented in Supplementary Table 1. As such, the purified enzyme should not contain cobalamin or A-cluster harboured by the ACDS subunits. The obtained complex should only catalyse the CO oxidation and F₄₂₀ reduction. Because the acetyl-CoA synthase was not part of this complex, CO/acetyl-CoA exchange is impossible and therefore was not attempted.

The main text has been modified as mentioned, and a sentence was added to the legend of Figure 1 to clarify the absence of β , γ , and δ subunits and, therefore, the absence of A-cluster and cobalamin.

Lines 875-878: “Purification steps of the CODH component of the ACDS on native PAGE (left). 1, soluble extract; 2, Anion exchange chromatography; 3, 4, hydrophobic exchange chromatography, and 5, size exclusion chromatography. The purified complex lacks the β , γ , and δ subunits and, therefore, does not harbour the A-cluster and B₁₂.”

Regarding the occupancy of the metal centers in the $\alpha_2\epsilon_2\zeta_2$ subcomplex, the electronic density and the b-factors of the modelled metalcenters do not suggest partial loss. Therefore, they have been modelled with a 100 % occupancy. This information is now added in the revised version:

Line 578-579: “Analysis of the b-factors of the cofactors indicates a 100 % occupancy in both models.”

5. There are no numbers on the many supplementary (Extended Data) Figures.

We assume the reviewer refers to figures present after the Extended Data Table 2 in the merged PDF file, as all other main figures, extended data figures and supplementary figures were appropriately numbered in the legend.

These are the original high-resolution figures in TIFF format provided during the submission that have not been downgraded during the automatic PDF conversion. They were provided for the reviewers to access high-resolution images and, therefore, are not numbered and do not have legends. Since all documents were carefully named with appropriate numbering, we had no alternatives. Please see the above comments for the revised version.

6. There are little biochemical data in the Main Figure section of the paper to support their conclusions and is limited to some activity measurements in Fig. 4 and Extended Data Table 1.

We agree with the referee that the enzymatic data are limited due to the extremely low available biomass. We concentrated our effort on extracting the maximum available information, but an exhaustive biochemical and enzymology characterisation of the enzyme would be impossible to perform from this scarce biomass. This limitation is a common issue in studying these enzymes, even when purified from pure cultures, and the published enzymatic characterisation is often relatively limited (e.g. Watanabe *et al.*, 2021; Nomura *et al.*, 2024). Please see our answer to the general comment for more details.

Still, we consumed the leftover of the biological material to provide additional enzymology data (presented in Extended Data Figure 6, see below).

The first column of “activity” should be labeled as Total Units.

The table was modified as proposed.

7. In Extended Data 2, the number of atoms presented should include a percentage completeness.

The percentage of completeness was added in the Extended Data Table 2.

8. After Extended Data 2, a bunch of Figures appear, including repeats of the main data figures – and some that have strange formatting and with no Figure legends. Presumably these are to be ignored??

Please see our response to point number 5.

9. Results, Line 66: “CO-oxidation activity carried by the CO-dehydrogenase subunit (CODH, α subunit)” – Isn’t the epsilon subunit required for CODH activity? Has the methanogenic alpha subunit ever been isolated alone?

The C-cluster, harboured by the α subunit, is the catalyst responsible for CO oxidation. To our knowledge, the α subunit from an archaeal system has not been isolated alone, as the ϵ subunit was systematically part of the archaeal CODH (sub)complexes. Therefore, assuming that the CO dehydrogenase module of archaea only contains the α subunit is incorrect, and we agree with the reviewer that the statement could have been misinterpreted. The reference to the α subunit was removed from the sentence:

Lines 73-74: “During the purification process, the ACDS was followed by measuring the viologen-dependent CO-oxidation activity of its CO-dehydrogenase (CODH) module.”

10. What is the ratio of the subunits based on densitometry scans of the SDS gel bands (relative to the molecular mass of the subunits).

Densitometry scans on denaturing gels suggest an equimolar subunit stoichiometry with a relative defect of the larger subunit (ACDS α or FwdA). The calculated ratios are $1 \pm 0/1.68 \pm 0.36/1.84 \pm 0.37$ (ACDS $\alpha/\zeta/\epsilon$) and $1 \pm 0/1.51 \pm 0.09/1.54 \pm 0.08/1.54 \pm 0.08/1.64 \pm 0.14$ (FwdA/B/I/C/G) when normalised on the calculated protein molecular weight and the signal of the larger subunit. This defect in the largest subunit could be explained by artefacts from the staining efficiency and SDS PAGE migration.

This experimental approach suffers from several putative limitations, staining artefacts, and biases, so we prefer not to include it in this work. The protein stoichiometry observed in the crystal structures ($\alpha_2\epsilon_2\zeta_2$ and FwdABCDGI) is, however, coherent with data obtained from native electrophoresis, size-exclusion chromatography (presenting a homogenous population) and mass spectrometry. Overall, these experiments show a coherent consensus on the quaternary organisation, affirming that the stoichiometry observed in the crystal structures is similar to that of the enzyme in solution.

11. Related to the title and throughout the paper, there is insufficient evidence supporting the concept that this ethanotroph reorients its overall metabolism towards F₄₂₀ reduction”, because they performed CO- and formylmethanofuran-dependent F₄₂₀ reduction capabilities of the two complexes and compared this to methyl viologen. Does they reduce ferredoxin? The key parameter related to use of F₄₂₀ as a substrate is the kcat/Km for F₄₂₀, compared to that with other electron acceptors (like ferredoxin, etc.) It is a large extrapolation to generalize this limited kinetic analysis even to making conclusions about the individual enzymes and an even larger leap to extend this to overall cellular metabolism.

We provide an argumentation regarding the “F₄₂₀ as a driver for ethanotrophy” model in the general comments and would like to stress some additional points.

Firstly, The F₄₂₀-reducing subunits concealed the ferredoxin-binding sites, limiting an efficient electron transfer to ferredoxin for catabolic reactions. Secondly, the measurement of CO:F₄₂₀ and FFA:F₄₂₀ oxidoreduction activities on isolated enzymes and in different cell extracts is in accordance with the postulated model that F₄₂₀ coupling is a specificity developed by ethanotrophs. Thirdly, the metagenome mining presented in Extended Data Fig. 8 also suggests an additional set of F₄₂₀-specialized enzymes compared to closely related archaea.

Nevertheless, we agree with the reviewer that the postulated “F₄₂₀ reoriented metabolism” is a model and not an observation because of the lack of experimental data for the other oxidoreductases operating the complete pathway. Therefore, we reformulated the text as follows:

Lines 23-27: “Based on these crystal structures, enzymatic activities, and metagenome mining, we propose a model in which the catabolic oxidising steps orchestrated by the ethanotroph would wire electron delivery to the F₄₂₀. Via this specific adaptation, the electron transfer from reduced F₄₂₀ to the sulphate-reducing partner would fuel energy conservation and represent the driving force of ethanotrophy”.

Lines 64-67: “The biochemical, structural and enzymatic characterisation of both complexes support that the CO₂-generating steps are coupled to F₄₂₀ reduction instead of ferredoxin, suggesting that F₄₂₀ reduction is the main driver of this metabolism²⁴.”

Lines 291-293: “In the proposed metabolic model (Fig. 5), the highly expressed Fpo complex (Supplementary Table 3) is the only energy-conserving system that would allow ions translocation across the membrane to fuel the ATP synthase.”

We measured additional activity with the extremely reduced available biological material to determine if both enzymes use ferredoxin as an electron acceptor. We compared different concentrations of F₄₂₀ and ferredoxin as electron donors (see Extended Data Figure 6 and the material and method section). These measurements showed that the F₄₂₀ extracted from *M. thermolithotrophicus* could be used as an electron acceptor with no apparent limitations in the enzyme affinity for the coenzyme. Catalysis with ferredoxin as an electron acceptor can be monitored, albeit with lower rates, in comparison to “classic” Fd-dependent CODH (Lemaire *et al.*, BioRxiv, <https://doi.org/10.1101/2024.07.29.605569>), the enzyme from *Ca. E. thermophilum* would exhibit a 260 times lower specific activity (137 U/mg of the CODH from *Clostridium autoethanogenum* versus 0.5 U/mg of the CODH from *Ca. E. thermophilum*). Therefore, if the reduction of ferredoxin is indeed possible, the reaction rates and the presence of additional F₄₂₀-reducing modules strongly argue towards a specificity towards F₄₂₀.

In light of these results and the presence of the additional F₄₂₀-reducing subunits, we conclude that the physiological acceptor *in vivo* is most probably the F₄₂₀.

We modified the text accordingly:

Lines 226-234: “Both enzymatic complexes also accept ferredoxin, albeit with reduction rates eightfold lower than those for F₄₂₀ (Extended Data Fig. 6). We interpreted that the difference in activity is due to the presence of the additional F₄₂₀-reducing subunits that hinder the access to the electron transfer chain by the ferredoxin. Together, our experimental data argue that F₄₂₀ is the most relevant electron acceptor of both complexes under physiological conditions. Side reactions have previously been measured for the bacterial ferredoxin-dependent CODH, in which flavodoxin or rubredoxin could be reduced³². However, the CODH from *Ca. E. thermophilum* is, to our knowledge, the first system depending on an additional reductase subunit to target a specific different electron acceptor.”

As discussed in the general comments, a more complete enzyme assay was impossible due to the extremely limited availability of biomass.

12. Line 81: “ACDS ... decompose in several populations on native PAGE, probably due to electrophoresis-dependent artefacts coming from a partial loss of cofactors or stabilising ions”. It is more likely due to the low affinity of the component subunits.

We agree with the reviewer. A low binding affinity may explain the native electrophoresis profile. Hence, we shortened the sentence in the main text.

Lines 111-113: “We suppose that the whole ACDS complex was destabilised upon cell lysis (e.g., due to the low affinity of the component subunits), simplifying the characterisation of the CODH component.”

13. Regarding the structures, the Redundancies, reflections, CC1/2s: (0.999 and 0.989 (0.621), and I/σI [10.8 and 7.2) values are good, and Rvalues seem fine, indicating no over-fitting.

We thank the referee for supporting our structural data.

14. Line 59: “questioning the dogma of the CO₂-releasing step coupled to ferredoxin in these organisms.” I question that the statement that CODH-ferredoxin is dogma. I’m sure that the authors can find some papers that study Fd as electron acceptor for CODH, however, a number of papers, starting from the earliest ones, point out how promiscuous CODH is in its electron carrier selectivity.

We agree with the reviewer that this sentence was not precise enough and could be misinterpreted due to the lack of metabolic and microbiological context. Here, we referred to ferredoxin as a physiological electron acceptor in ACDS and Fmd/Fwd complexes from methanogens and alkanotrophs. The current assumption on these complexes

is that ferredoxin is the physiological electron shuttle, a concept systematically used when drawing metabolic pathways of “omics” studies. Unfortunately, this dogma in methanogens and related alkanotrophs was never questioned, and we hope that our study will help the community to open possibilities of alternative electron donors/acceptors.

The sentence was modified.

Lines 53-56: “The ethanotrophs do not contain any known membranous systems that would allow energy conservation from ferredoxin oxidation, questioning if the CO₂-releasing step operated by ACDS and Fwd/Fmd would be necessarily coupled to ferredoxin reduction, an established dogma in methanogens and alkanotrophs.”

We also reinforced the statement that such a hypothetical pathway was first drawn from omics data based on the methanogenic pathway.

Lines 40-42: “It has been suggested that the generated ethyl-CoM is further processed to acetyl-CoA based on the knowledge acquired on methanogens belonging to the same order, together and supported by transcriptomics and proteomics data^{7,8}.”

Furthermore, CODH has been shown to reduce both flavodoxin and flavins, so it is not surprising that it also can reduce F420, a deazaflavin. Perhaps this should be cited within the paper.

We now acknowledge that bacterial CODHs use a wider range of electron acceptors.

Lines 230-234: “Side reactions have previously been measured for the bacterial ferredoxin-dependent CODH, in which flavodoxin or rubredoxin could be reduced³². However, the CODH from *Ca. E. thermophilum* is, to our knowledge, the first system depending on an additional reductase subunit to target a specific different electron acceptor.”

Composition issues:

1. Abstract: “methanogenic or methanotrophic relative”. Please clarify the meaning of this. It seems unlikely that the CODH and formate dehydrogenases in ethanotrophic organisms are so different in their ability to reduce F420, that the methanotrophic and methanogenic enzymes are unable to reduce F420. It would be very interesting if this is true and unambiguously demonstrated.

The previous sentence was not correctly formulated because the information on the cell extract experiment and metagenome analyses was lacking.

Our analyses indicate that the genomes of methanogens and ANME globally do not encode the F₄₂₀-reducing modules connected to ACDS ζ and FwdI subunits (Extended data Figure 7 and 8). More importantly, our activity measurements could not detect any CO:F₄₂₀ and FFA:F₄₂₀ oxidoreductase activity in the extracts of *Methanosarcina barkeri* or ANMEs (Fig. 4). Moreover, previously characterised ACDS subcomplex (Gong *et al.*, 2008) or Fwd/Fmd complexes (See Wagner, Ermler and Shima, 2018 and Watanabe *et al.*, 2021) of methanogenic species do not contain directly connected F₄₂₀ reducing modules comparable to those of the complexes of *Ca. E. thermophilum*.

To emphasise this statement, we modified the abstract:

Lines 21-27: “Accordingly, both systems exhibit robust F₄₂₀-reductase activities, which are not detected in the cell extract of related methanogens and anaerobic methane oxidisers. Based on these crystal structures, enzymatic activities, and metagenome mining, we propose a model in which the catabolic oxidising steps orchestrated by the ethanotroph would wire electron delivery to the F₄₂₀. Via this specific adaptation, the electron transfer from reduced F₄₂₀ to the sulphate-reducing partner would fuel energy conservation and represent the driving force of ethanotrophy”.

2. Abstract: “coupling C1-oxidation”. Be specific about what compounds are being oxidized here, formate and CO.

The abstract was modified as proposed.

Lines 17-21: “While it has been assumed that the CO₂-forming CO-dehydrogenase and formylmethanofuran-dehydrogenase are coupled to ferredoxin reduction in homologues, we found that both reactions deliver electrons to the F₄₂₀ cofactor in the *Ca. E. thermophilum*. Both multi-metalloenzyme complexes harbour electronic bridges connecting CO and formylmethanofuran oxidation centres to a bound flavin-dependent F₄₂₀-reductase.”

3. Intro: “Only two archaeal species belonging to the Methanosarcinales order can catalyse the complete anaerobic oxidation of ethane”. This is ambiguous. Do the authors mean that there are only two archaeal species (no others exist on earth) that catalyze ethane oxidation? Or that only two species have been isolated so far that catalyze ethane oxidation and they both are Methanosarcina?

Anaerobic ethane oxidation has only been demonstrated in two organisms. Both of them belong to the archaeal *Methanosarcinales* order. The sentence was modified for clarification.

Lines 35-37: “The two ethane oxidisers are part of the *Methanosarcinales* order and were shown to catalyse the complete anaerobic oxidation of ethane, the second most abundant alkane in seeps⁷⁻⁹.”

4. Intro (line 46-7): “Therefore, ethane would be ultimately converted into two molecules of CO₂ and CO₂-releasing steps (ACDS and Fwd/Fmd complexes) would be coupled to ferredoxin reduction,” As written, this also is ambiguous/incorrect, because grammatically it indicates that “CO₂-releasing steps (ACDS and Fwd/Fmd complexes) would be coupled to ferredoxin reduction”. Please clarify. It would probably be more correct to state that “Therefore, ethane oxidation (involving/or including ACDS and Fwd/Fmd complexes) to CO₂ would be coupled to ferredoxin reduction”.

The sentence has been modified as the reviewer proposed.

Lines 48-50: “Therefore, ethane would be ultimately oxidised into two molecules of CO₂, and the CO₂-releasing enzymes (ACDS and Fwd/Fmd complexes) are expected to reduce ferredoxin, which is employed for energy conservation in methanogens¹⁵.”

Reviewer #2 (Remarks to the Author):

The manuscript by Lemaire et al, utilises native purification and x-ray crystallography to structurally characterise the CODH and Fwd complexes from the *Achaea Candidatus Ethanoperedens thermophilum*, grown in an enrichment culture. *Candidatus Ethanoperedens thermophilum* is one of the few microbes capable of the anaerobic oxidation of ethane, which it performs syntrophically with a sulphate reducing bacteria. The CODH and Fwd complexes are thought to play key a role in this process, through the oxidation of carbon monoxide and formyl-MFR.

The manuscript is well written and comprehensively describes the detailed structural characterisation of these complexes, which is backed up by functional and bioinformatic analysis. The authors approach to obtaining these complexes is innovative, and the challenge of natively isolating these complexes from anaerobic enrichment cultures cannot be overstated. The authors should be commended for their technical expertise in producing a sample that is high quality enough to generate crystals that diffract to high resolution.

We sincerely thank the reviewer for the time invested in the reviewing process and the constructive comments.

The key finding of the study is that both the CODH and Fwd complexes contain a module homologous to a F420-reductase, strongly implying that these complexes donate electrons from CO and formyl-MF to the redox cofactor F420. The authors support this with activity assays of the purified complexes and cell lysates showing that CODH, Fwd and cell lysates from the Ethane50 enrichment culture can use F420 as an electron acceptor, while cell lysates from ANME-1/2 and *M. barkeri* cannot.

The authors then propose a model based on these data where *Candidatus Ethanoperedens thermophilum* use F420 as their main electron acceptor, which has the advantage of making ethane oxidation exergonic, creating favourable thermodynamic conditions for its oxidation.

Overall, the study is well conducted, and the structural data is compelling. However, I think it is necessary that the authors address the relatively low reaction rates observed for CODH, Fwd and Ethane50, when F420 is utilised as the electron acceptor. For the purified enzymes these rates are significantly lower than the non-specific electron acceptor methyl viologen. It isn't intuitive to me why this would be the case if F420 is the physiological electron acceptor substrate for the enzymes.

Indeed, we did not discuss the higher activity with MV as an electron acceptor instead of F₄₂₀ for different reasons.

Firstly, the difference between a surrogate viologen dye and the physiological electron acceptor F₄₂₀ has already been observed for the F₄₂₀H₂-oxidising sulphite reductase from *Methanocaldococcus jannaschii* (Johnson and Mukhopadhyay, 2005). In this study, authors measured a specific activity three times higher with reduced MV than reduced F₄₂₀ (32 μmol of electron transferred/min/mg for F₄₂₀H₂ and 90 μmol of electrons transferred/min/mg protein for reduced MV). Such observation can be rationalised by the fact that viologen dye could shortcut the electron path and directly uptake the electrons from the oxidase catalytic centre, bypassing the F₄₂₀-reductase module. Hence, the F₄₂₀ reducing module may be the limiting step of the overall reaction catalysed by the enzymes under these experimental conditions.

Secondly, the concentration of electron acceptors used for the activity measurements differed drastically (5000 μM for MV and 20 μM for F₄₂₀). Therefore, the specific activities were hardly comparable as the concentration of F₄₂₀ (kept low due to the relatively low yield of the cofactor purification from the methanogen *M. thermolithotrophicus*) may not be sufficient to ensure a maximal turnover. To verify if the use of F₄₂₀ concentration was below the K_m, we performed activities at 20, 30 and 50 μM concentrations. The additional data provided in Extended Data Fig. 6 shows no clear sign of a low affinity for F₄₂₀ for both enzymes, confirming that the used F₄₂₀ concentration seems sufficient for suitable activity measurements.

Finally, the F₄₂₀ variant purified from the *Methanococcales Methanothermococcus thermolithotrophicus* might differ slightly from the native F₄₂₀ from *Ca. E. thermophilum*. These putative differences would unlikely affect activities in light of the experiment provided in Extended Data Figure 6 (i.e., as the increase in F₄₂₀ concentration did not significantly affect the rates). However, we cannot rule out the possibility that it may result in lower turnover rates. This point was discussed in the text.

Lines 215-225, the modified part being underlined: “The measured rates of F₄₂₀ reduction in both complexes are lower than that of the methyl-viologen reduction. This difference has similarly been observed in the F₄₂₀H₂-oxidising sulphite reductase from *M. jannaschii*³¹. Here, it suggests that under these experimental conditions, the F₄₂₀-reductase module is the rate-limiting step compared to the substrate oxidation rates. In that case, methyl-viologen reduction rates will be higher because this artificial electron acceptor could directly uptake electrons near the substrate oxidative centre. It must also be noted that the F₄₂₀ used in this study was extracted from the *Methanococcales Methanothermococcus thermolithotrophicus*. Differences in the coenzyme structure compared to the native F₄₂₀ from *Ca. E. thermophilum* may limit the reaction kinetics. However, activity measurements with different F₄₂₀ concentrations did not suggest a low affinity for the coenzyme (Extended Data Fig. 6).”

Obviously, it would be great to obtain a crystal structure of CODH and Fwd in complex with F420 or a derivative, but I realise this may not be a reasonable request given the amount of data already in the study. However, molecular docking and/or structural analysis (given the negatively charged glutamate tail of F420, there is likely to be a corresponding positively charged region at the acceptor substrate binding site) to support F420 binding would be appropriate.

Resolving a F₄₂₀-bound structure for such a type of F₄₂₀-reductase would indeed be a challenging task. Extensive efforts have been made in that regard for the F₄₂₀H₂-oxidising sulphite reductase (Jespersen et al., 2022) and the F₄₂₀-reducing hydrogenase (Allegretti et al., 2014; Vitt *et al.*, 2014) without any success. Given the extremely limited available amounts of biological material, we did not attempt to obtain the F₄₂₀-bound complex.

We followed the referee’s advice of presenting the electrostatic surface of the F₄₂₀ reductase module to illustrate the conserved binding site for the polyglutamate tail. As expected, the F₄₂₀-reducing modules of both enzymatic complexes exhibit positively charged patches around the F₄₂₀ reduction site similar to other characterised F₄₂₀-reductases. The structural analysis was added as Supplementary Fig. 9 and detailed in the text.

Lines 210-213, the modified part being underlined: “The access to the catalytic FAD in the ACDS ζ subunit and FwdI is surrounded by a positively charged surface that would stabilise the polyglutamate group of the F₄₂₀ tail, as proposed in other F₄₂₀-dependent enzymes (Supplementary Fig. 9).”

Additionally, further activity measurements in the presence of additional acceptor substrates could further clarify the physiological electron acceptor for these enzymes.

Following the reviewer’s suggestion, we used the leftovers of the biological material to perform activity measurements with different concentrations of F₄₂₀ and ferredoxin to assess the selectivity of the enzymes for the electron acceptors. These additional data are presented in Extended Data Fig. 6. Shortly, catalysis with ferredoxin as an electron acceptor can be monitored, albeit with lower rates, in comparison to “classic” Fd-dependent CODH (Lemaire *et al.*, BioRxiv, <https://doi.org/10.1101/2024.07.29.605569>), the enzyme from *Ca. E. thermophilum* would exhibit a 260 times lower specific activity (137 U/mg of the CODH from *Clostridium autoethanogenum* versus 0.5 U/mg of the CODH from *Ca. E. thermophilum*). Therefore, we conclude that if the reduction of ferredoxin is indeed possible, the reaction rates and the presence of additional F₄₂₀-reducing modules strongly argue towards a specificity towards F₄₂₀.

Line 226-234: “Both enzymatic complexes also accept ferredoxin, albeit with reduction rates eightfold lower than those for F₄₂₀ (Extended Data Fig. 6). We interpreted that the difference in activity is due to the presence of the additional F₄₂₀-reducing subunits that hinder the access to the electron transfer chain by the ferredoxin. Together, our experimental data argue that F₄₂₀ is the most relevant electron acceptor of both complexes under physiological conditions. Side reactions have previously been measured for the bacterial ferredoxin-dependent CODH, in which flavodoxin or rubredoxin could be reduced³². However, the CODH from *Ca. E. thermophilum* is, to our knowledge, the first system depending on an additional reductase subunit to target a specific different electron acceptor.”

Point-by-point response to the reviewers' comments

REVIEWERS' COMMENTS

Reviewer #1 (Remarks to the Author):

The manuscript by Lemaire et al, describes predominantly x-ray crystallographic analysis to reveal the structure of the CODH and Fwd complexes from the *Achaea Candidatus* *Ethanoperedens thermophilum*. This is one of the few microbes capable of the anaerobic ethane oxidation, which it performs syntrophically with a sulphate reducing bacteria. The CODH and Fwd complexes play key a role in ethane oxidation, through the oxidation of carbon monoxide and formyl-MFR. The manuscript comprehensively describes the detailed structural characterisation of these complexes, which is backed up by some functional and bioinformatic analysis. The authors should be commended for meeting the challenge of natively isolating these complexes from anaerobic enrichment cultures and producing a sample that is high quality enough to generate crystals that diffract to high resolution. The revision is improved. There are just a few additional changes in toning down some of the hyperbolic language in favor of perhaps more accurate language. Otherwise, I think it's a useful contribution to the literature.

We thank the reviewer for the time and interest given to our work. The text was modified according to the reviewer's suggestions, with explanations and modifications detailed below.

1. The authors continue to state that they : “broke the long-standing dogma” that CO₂-forming reactions in the acetyl-CoA reducing pathways are coupled to ferredoxin.) No. It has been demonstrated that CODH is promiscuous in its redox partner choice. What has been known since CODH was first isolated is that this enzyme has a very wide range of electron acceptors. The actual electron acceptor in the case of F₄₂₀ appears to be a 4Fe-4S metal center in the F₄₂₀ reductase. They need to compare Fd and F₄₂₀ as electron acceptors in terms of CO oxidation to validate the point that the CODH from ethane-oxidizing organisms have a special redox property (“Accordingly, both systems exhibit robust coupled F₄₂₀-reductase activities, which are not detected in the cell extract of related methanogens and anaerobic methane oxidisers.”

The reference to the ferredoxin as electron carrier “dogma” was centred on the CO₂-releasing reaction of the reductive acetyl-CoA pathway in the archaeal metabolism and, more precisely, on the Fwd/Fmd and ACDS systems found in methanogens and alkanotrophs. These reactions are systematically assumed to depend on ferredoxin without experimental validation.

We removed the reference to any “dogma” and reformulated the sentence:

Lines 59-62, the modified part being underlined:

“The ethanotrophs do not contain any known membranous systems that would allow energy conservation from ferredoxin oxidation, questioning if the CO₂-releasing step operated by ACDS and Fwd/Fmd would be necessarily coupled to ferredoxin reduction, as it is commonly assumed for methanogens and alkanotrophs.”

We understand the point of the reviewer that the physical electron donor of the CODH defined as the α/ϵ subunits is the distal [4Fe-4S]-cluster of the F₄₂₀-reductase (ζ -subunit). However, the CODH native complex is referred to as the purified $\alpha_2\epsilon_2\zeta_2$ complex (CODH component). Here, the final electron acceptor of the CODH component is the F₄₂₀, which is reduced by the ζ subunit thanks to the catalytic

FAD cofactor. The FAD is electronically connected via a [4Fe-4S] clusters network to the C-cluster, the catalyst of the CO-oxidation concealed in the α subunit. Hence, in this context, the electron acceptor of the “CODH component” is F_{420} and not the first cluster of the ζ subunit.

This was stressed in the text to avoid confusion, and provide an easier access to the concept for a broader audience (such as microbiologists). Therefore, we verified that the complex is systematically referred to as “CODH component” or “CODH subcomplex” in the text, and not simply “CODH”, which would indeed be misleading in that sense.

Lines 107-110, the added part being underlined:

“Because of its tight binding on the CODH component and its presence in the operon coding for the ACDS, this additional subunit will be referred to as ζ subunit (CAD7772047). Unless stated otherwise, the name CODH component will systematically refer to the $\alpha_2\varepsilon_2\zeta_2$ complex below.”

The consequence of the $\alpha_2\varepsilon_2\zeta_2$ organization is that the distal [4Fe-4S] cluster of the CODH α -subunit (cluster 4) would not be solvent-accessible for a putative electron delivery to a ferredoxin. The comparison of the rates of ferredoxin and F_{420} reduction, added as additional experiments following the recommendation of both reviewers (Supplementary Fig. 15), also confirms the proposal of F_{420} as the physiological electron acceptor. Hence, we propose that F_{420} is the physiological electron acceptor of the overall ACDS reaction (as the CODH component would be part of the ACDS), even if a minor contribution to the reduction of the ferredoxin pool cannot be excluded, as discussed in the manuscript.

Furthermore, is incorrectly stated that CODH-dependent reduction of flavodoxin and rubredoxin is a “side reaction”. The cited study looked at various redox partners including ferredoxin, flavodoxin and rubredoxin alone. They all accept electrons from CODH. It should be made clear that the actual electron acceptor for CODH is an FeS cluster in the F_{420} reductase, which reduces the F_{420} .

As stated above, the electron acceptor of the α subunit harbouring the C-cluster is indeed the distal [4Fe-4S] cluster of the ζ subunit. However, the electron acceptor of the CODH component ($\alpha_2\varepsilon_2\zeta_2$ complex) is F_{420} . The CODH subcomplex is proposed as part of the ACDS for different reasons stated in the main text.

Nevertheless, we now stress that point even stronger:

Lines 165-168, the added part being underlined: “The anchoring of the ζ subunit to the $\alpha_2\varepsilon_2$ core through its N-terminal domain (Fig. 2d) offers a structural template to picture how the electrons from the distal cluster of the α subunit (cluster 4) will be transferred on the distal cluster of the ζ subunit (cluster 5). The structure also depicts how a soluble ferredoxin would dock on ferredoxin-dependent archaeal ACDS.”

While the reviewer's statement of the ζ subunit being the electron acceptor of the CODH protein (so the α/ε subunits) is legitimate in terms of uncoupling the enzymatic reaction, it has little direct implications on the organism's metabolism. In other words, instead of considering the complex as two distinct entities: (1) A CODH reducing a F_{420} -reductase and (2) a F_{420} -reductase reducing F_{420} , we present the complex as one: a CODH component reducing F_{420} , which is scientifically correct. The two other systems that have described an oxidoreductase directly coupled to a F_{420} -reductase (F_{420} -reducing hydrogenase, Vitt et al. J. Mol. Biol. 2014 and the F_{420} -dependent sulfite reductase, Jespersen et al. Nat. Chem. Biol. 2023) were also described as a single entity.

We verified that the text always refers to the “CODH component” (and not simply the CODH) when mentioning the CO: F_{420} oxidoreductase complex.

Regarding the qualifications of CODH side reaction in which electrons are transferred to alternative acceptors, we agree that the cited study tested and confirmed the reduction of rubredoxin and flavodoxin

in vitro. In our work, we were not referring to the reduction of these molecules as “side reactions” in terms of enzymatic assay but in terms of the physiological context. This context was set to avoid restricting the story to the niche of anaerobic biochemistry but extend it to microbiologists.

Indeed, decades of work on anaerobes, from which the CODHs used in the study were extracted, did not highlight rubredoxin as part of the central metabolism of acetogenic bacteria. In the vast majority of the studied biological systems, ferredoxin appears to be the physiological electron donor or acceptor of the CODH-containing enzymes. Nevertheless, we also understand the argument of the referee, in particular regarding the use of flavodoxins, and modified the sentence to clarify this point.

Lines 237-242, the modified part being underlined:

“Together, our experimental data argue that F_{420} is the most relevant electron acceptor of both complexes under physiological conditions. It must be noted that alternative electron acceptors (e.g., flavodoxin) have already been described for purified bacterial ferredoxin-dependent CODH³². However, the CODH from *Ca. E. thermophilum* is, to our knowledge, the first system depending on an additional reductase subunit to target a specific different electron acceptor.”

2. The manuscript should include color coding on each figure to identify the proteins and subunits.

The manuscript uses similar colour coding for all figures, as indicated in the figures and the legends of every main and supplementary figure.

3. While they have partly clarified the nature of the protein (CODH, not ACDS), they continue to state that they performed an enzymatic characterization (instead of performing an enzyme assay).

The text was modified:

Lines 70-73, the modified parts being underlined:

“The biochemical and structural characterisation of both complexes, as well as enzymatic assays, support that the CO_2 -generating steps are coupled to F_{420} reduction instead of ferredoxin, suggesting that F_{420} reduction is the main driver of this metabolism²⁴.”

In terms of nomenclature, there are still places where they talk about ACDS instead of CODH. For example, it is not ACDS that is CO_2 releasing and especially in their complex, it is CODH.

The text was carefully checked, especially regarding the references to both the CODH component (the purified enzyme, $\alpha_2\varepsilon_2\zeta_2$) and the full ACDS complex (composed of the $\alpha_2\varepsilon_2\zeta_2$ subcomplex and the other subunits, in a yet unknown stoichiometry in this organism).

While the context was previously clearly established (Lines 79-80: “During the purification process, the ACDS was followed by measuring the viologen-dependent CO-oxidation activity of its **CO-dehydrogenase (CODH) module**”), we emphasised even more the fact that $\alpha_2\varepsilon_2\zeta_2$ subcomplex must be part of the ACDS supercomplex.

Lines 131-134, where the addition is underlined: “All metallo-cofactors harboured on the α subunit part of the ACDS of *Ca. E. thermophilum* are coordinated by residues similar to those described in the *M. barkeri* homologue, even if some substitutions in the close environments of the cofactors might tune their redox potentials (Supplementary Fig. 6).”

F_{420} reduction by the complete ACDS ($Acetyl-CoA + F_{420} \rightarrow CoA + methyl-tetrahydromethanopterin + CO_2 + F_{420}H_2$) is a hypothesis based on the data gathered on the subcomplex. For instance, lines 276-277: “...the F_{420} -reduction coupled to substrate oxidation catalysed by ACDS and Fwd **would** be highly exergonic.”

Our reasons to consider the CODH component as a part of the ACDS supercomplex are clearly stated (lines 113-120). The conclusions of the experiments carried out on the purified $\alpha_2\epsilon_2\zeta_2$ subcomplex hence have implications on the reaction performed by the full ACDS. As an example, our results show a F_{420} reduction through CO oxidation (releasing CO_2) by the $\alpha_2\epsilon_2\zeta_2$ subcomplex, which implicates that the complete ACDS, containing the $\alpha_2\epsilon_2\zeta_2$ subcomplex, would reduce F_{420} from acetyl-CoA, through the intermediate generation of CO at the A-cluster (and the release of CO_2 and methyl-tetrahydromethanoptin or a similar molecule).

This discussion is on the same line as previous remarks considering enzymatic complexes considered as multiple entities, with each of them generating a single isolated reaction (i.e., **1 [Acetyl-CoA decarboxylase]**. Acetyl-CoA \rightarrow CoA + methyl- B_{12} + CO; **2 [Methyltransferase]**. Methyl- B_{12} + tetrahydromethanoptin \rightarrow B_{12} + methyl-tetrahydromethanoptin; **3 [CODH]**. CO + H_2O + oxidised F_{420} -reductase \rightarrow CO_2 + $2H^+$ + reduced F_{420} -reductase; **4 [F_{420} reductase]**. Reduced F_{420} -reductase + $2H^+$ + F_{420} \rightarrow oxidized F_{420} -reductase + $F_{420}H_2$), or one entity performing the overall reaction (i.e., **[ACDS]** Acetyl-CoA + F_{420} \rightarrow CoA + methyl-tetrahydromethanoptin + CO_2 + $F_{420}H_2$). We preferred the latter as it is clearer to illustrate catabolism, in particular in the field of microbiology.

Reviewer #2 (Remarks to the Author):

The authors have responded in detail to my comments/suggestions regarding the manuscript, and implemented appropriate changes to the manuscript to address them. This is a very nice and well rounded study. I don't have any further comments or concerns.

We thank the reviewer for the time and interest given to our work.